

# Econometric Modelling for Estimating Direct Flood Damage to Firms: A Local-Scale Approach Using Post-Event Records in Italy

Marta Ballocci[1,2], Daniela Molinari[1], Giovanni Marin[3], Marta Galliani[1], Alessio Domeneghetti[4], Giovanni Menduni[1], Simone Sterlacchini[5], and Francesco Ballio[1]

[1]Dept. of Civil and Environmental Engineering DICA, Politecnico di Milano, Milano, Italy
[2] University School for Advanced studies, Pavia, Italy.
[3]Department of Economics, Society, Politics, University of Urbino 'Carlo Bo', via Aurelio Saffi, 2, 61029 Urbino, Italy
[4]Dept. of Civil, Chemical, Environmental and Materials Engineering, Alma Mater Studiorum - University of Bologna, Bologna, Italy
[5]Institute of Environmental Geology & Geoengineering, National Research Council of Italy, Milano, Italy.

*Correspondence to*: Marta Ballocci (marta.ballocci@mail.polimi.it)

Keywords: Flood risk; Damage assessment; Disaster impact; Economic activities; Mitigation; Forecast model.

**Abstract.** Managing flood risk is crucial for achieving global sustainability. Particularly, flood damage to firms' assets imposes significant financial stress, necessitating efforts to minimize future consequences. However, current tools and knowledge for estimating flood damage to firms are inadequate, primarily due to a lack of high-quality damage data and the diversity of enterprise characteristics, complicating generalization. This study aims to improve understanding micro-local scale flood damage to firms in Italy through the analysis of empirical data, focusing specifically on direct damage. The dataset comprises 812 observed damage records collected after five flood events. Damage is categorized into building structure, stock, and equipment. The analysis reveals relationships between damage, economic sector, and water depth. Results indicate that damage increases at a rate less than proportional to the firm surface area and with water depth significantly explaining only stock damage. The quantification of the damage across different sectors highlights, healthcare, commercial, and manufacturing categories as the most vulnerable for building structure, stock, and equipment damage, respectively. The derived damage model offers better predictive accuracy than foreign models in the Italian context. These findings aid in developing effective, tailored risk mitigation strategies and provide valuable insights for future research and policy aimed at reducing flood impacts on firm in Italy.




## 1 Introduction

This paper presents an empirical analysis of flood damage data related to Italian economic activities, aimed at developing a
forecast model to estimate microscale direct flood damages to Italian enterprises.

Flood risk is one of the main concerns of policymakers aiming for a more sustainable future (see, e.g., the Sendai Framework for Disaster Risk Reduction, the UN Sustainable Development Goals, and the European Floods Directive). Europe has 18.7 % of its territory exposed to high flood hazard (Arrighi et al., 2013), while Italy has, respectively, 5.4%, 10% and 14% of its territory exposed to hydro-geological risk in the high, medium and low probability hazard scenario (ISPRA, 2021). Projected
climate change scenarios indicate a rise in precipitation intensity that, without proper adaptation policies and measures, will results in higher flood risk with significant negative consequences (IPCC, 2022).

Managing flood risk is essential for protecting people and properties, preserving the environment, and minimizing economic impacts of floods on societies. Especially, damage to business is a significant source of financial stress after floods. Reduced sales during and in the aftermath of the event, damaged stock, and disrupted equipment and machinery all affect business
interruption and pose challenges to recovery, especially for uninsured or financially strained enterprises (Samantha, 2018).

The study of flood impacts on the different sectors that compose the built environment, and society is crucial to improve the information used by modelers and decision-makers to guide flood risk management. This includes prevention, protection, mitigation and risk-aware planning actions to reduce potential adverse consequences of floods (Bremond et al., 2018). In particular, the analysis of direct damages to economic activities provides insight into the quantification of costs across
economic sectors and the interaction between damage and its explanatory variable (like hazard intensity and the characteristics of different economic activities) thereby enabling the implementation of adaptation measures to reduce avoidable costs. The assessment of damages is important to evaluate actions from the individual entrepreneur to the public decision-makers, enabling public or private preventive policies against damages. Knowledge and understanding of the specificities of each case permit to implement tailored measures to a specific economic sector, while preparedness is becoming increasingly important
in managing the magnitude of impact that can be mitigated by population behaviour, incentivizing insurance policies and adaptive measures (Balbi et al., 2013).

Despite the business sector assumes a critical role, for both its importance for the economic well-being of society and the high losses it suffers in case of inundations, methods to assess damage to economic activities are much less developed and affected by higher uncertainty compared to other sectors, such as the residential (Gissing and Blong, 2004; Samantha, 2018; Zhou and
Botzen, 2021). To provide an overview of the extant literature, Table 1 displays the number of papers retrieved from the Scopus database through queries conducted on the title, abstract, and keywords.

In fact, studying the damage to the business sector means facing a complex problem, involving interconnections among several systems (e.g., society, the reference market, the financial system) as floods may have devastating effects not only on business survival but also on the economic and social fabric (Menoni et al., 2016; Wedawatta et al., 2014).





Damage to economic activities is generally distinguished in three different impacts: direct damages to capital goods, including moveable and unmoveable goods (structure of the building, equipment, stock), caused by contact with water; damages caused by business interruption to the affected economic activities inside the flooded area; indirect damages or costs suffered by business in the supply chain outside the flooded area, deriving from the supply interruption from businesses in the flooded area (Burzel et al., 2018). The three impacts are strongly interconnected with each other.

**Table 1.** Number of papers in Scopus by keywords in the title, abstract and keywords (LIMIT-TO (DOCTYPE, "ar")) AND (LIMIT-TO (LANGUAGE, "English")) AND (LIMIT-TO (EXACTKEYWORD, "Flood Damage")). (Last access: May 2024).

| Strings | Number of papers |
| --- | --- |
| Flood* AND damage* AND firm*OR enterprise* | 100 |
| Flood* AND damage* AND residential OR building* | 172 |
| Flodd* AND damage* AND crop* OR agriculture | 211 |
| Flood* AND damage* AND infrastructure | 716 |
| Flood* AND damage* | 2907 |

The analysis presented in this paper contributes to flood damage modelling for economic activities by proposing a model for the estimation of direct damage. More specifically, the aim of this work is to enhance modelling capability and knowledge of damage mechanisms, towards more reliable and comprehensive flood risk assessment.

The analysis is conducted by investigating empirical damage data collected in the aftermath of flood events through an econometric approach with a multiple regression model. Data refer to different floods that affected Italy in the last twenty years, all characterized by the common feature of being riverine and low-velocity floods. The analysis goes beyond considering water depth as the sole explanatory variable for damage. In fact, previous studies (Van Ootegem et al., 2015; Wagenaar et al., 2017; Endendijk et al., 2022) have utilized regression analysis to integrate non-hazard indicators, recognizing the importance

of considering additional factors beyond water depth in understanding and modelling flood risk.

The analysis is carried out at the micro-scale (i.e., at the single economic activity level), (i) studying the relationship between damage (total, and subdivided by components: structure, equipment, stock), and their main explanatory variables (water depth, business size, economic sector), and (ii) obtaining a preliminary estimation of flood damage for different categories of activity. The results provide a forecast model for the total damage and a preliminary model for the single damage components. The

prediction offers a clearer idea of which variables are statistically significant in determining differences in the amount of damage suffered by business due to a flood event in Italy at a local scale.

The paper is organized as follows: Section 2 provides an overview of available damage models for the business sector in Europe and Italy, along with a discussion of their limitations, emphasizing the need for the current research. Sections 3 describes the case studies and data used in this analysis. Section 4 presents the methodology underpinning the analysis. Section

5 and 6 detail the results along with model validation. Section 7 compares the performance of existing models with the model



proposed in this study, highlighting improvements over the current state of the art. Section 8 discusses the strengths and limitations of the present work, along with directions for further research. Section 9 provides concluding remarks.

## 2 Flood damage to firms: state-of -the-art

The estimation of damages to economic activities includes both direct and indirect damage. Direct damages are usually assessed using a damage function that depends on hazard, vulnerability and exposure (Koks et al., 2015). On the other hand, indirect damages are commonly assessed using General Equilibrium Models or Input-Output models, which measure how the effects of a disaster propagate throughout the economy over a period of time (Allaire, 2018). However, indirect flood damage assessment is beyond the scope of this analysis.

With respect to direct damages, two main approaches are commonly used: empirical estimations and synthetic models. Empirical estimations are typically based on data collected from past flood events, while a synthetic model uses assumptions about damage mechanisms (Dottori et al., 2016). Most past studies on damage to the business have utilized empirical models. Despite the scarcity of observed damage data, which makes calibration and validation problematic (especially for contexts that differ from the original case study), empirical models are usually preferred when assessing business damage. The wide heterogeneity of economic activities makes it difficult to analyse all damage mechanisms using a synthetic approach.

Examples of micro-scale models for assessing direct flood damage to economic activities can be found both in Europe and Italy (Amadio et al., 2016; Arrighi et al., 2013; Grelot and Richert, 2020; Kreibich et al., 2007; Martinez et al., 2018; Martínez-Gomariz et al., 2020; Penning-Rowsell, 2005), or outside Europe (Bari et al., 2021; Hasanzadeh Nafari et al., 2016; Hu et al., 2019; Hung and Diep, 2022; Kuroda et al., 2022; Li et al., 2018; Olmez and Deniz, 2023; Samantha, 2018; Scawthorn et al., 2006; Wedawatta et al., 2014).

In the European context, FLEMOcs (Kreibich et al., 2010) is an empirical model based on data collected in Germany, which estimates the loss ratio of buildings, equipment, goods, products and stock for four sectors: public and private services, industry, corporate services and trade. According to the model, losses depend on several hazard and vulnerability variables: water depth, economic sector, company size, precautionary behaviour taken by company owners and the level of contamination of water. In the UK, the Multicoloured Manual (Penning-Rowsell et al., 2005), includes synthetic damage functions to estimate the damage to different categories of non-residential properties, including economic activities (e.g., retail, office, distribution/logistics, manufacturing). Susceptibility functions proposed by the model relate damage to water depth, flood duration and business surface. France developed national damage functions to assess damage to equipment, stock, and structure of economic activities, as a function of water depth and flood duration (Bremond et al., 2018.; Grelot and Richert, 2019). In Spain, Martínez-Gomariz et al., (2020) developed national empirical flood depth-damage curved, based on actual data in Barcelona, for buildings, stock and inventories for the following economic sectors: warehouses, car parks, restaurants, general





trading, homeowners associations, sport, education, hotels, industries, offices, health, workshops, dwellings, churches & singular buildings. Functions vary with the category of the activity, identified by the NACE code.

Consensus within literature highlights the importance of being cautious when attempting to transfer damage models within different countries, as it involves a notable degree of uncertainty, particularly within regions lacking sufficient data. Moreover,
it is recommended to use models developed in regions similar to that of the initial application, because characteristics of floods, vulnerability or exposed elements and relative values are strongly context specific (Cammerer et al., 2013; Merz et al., 2010; Smith, 1994). This makes the use of foreign models in the Italian context very difficult due to scarcity of data for their validation (see Section 7).

Two main studies in Italy propose methods to assess the potential damage to economic activities. Arrighi et al. (2013) propose
a method to assess direct damage to structures and contents of commercial activities at the scale of the census block, i.e. at the meso-scale, by developing stage-damage curves for the urban context of Florence. However, this model is strictly related to the exposure and vulnerability parameters of the city of Florence, therefore hardly transferable to a different context. Molinari et al. (2016) develop another approach, named Flood-IMPAT, to assess direct damage to the business sector, again at the meso-scale. They propose the net capital stock, supplied by the Italian Institute of Statistics (ISTAT), as a proxy indicator of the
value of contents and structure of businesses and uses the depth-damage functions by the International Commission for the Protection of the Rhine – ICPR (ICPR 2001) to assess the damage, distinguishing by economic sector using the NACE code. Still, the authors highlight the limitation of using a foreign model, in terms of reliability of results, and the importance of implementing models that are specific for the context under investigation and that work at lower scales. Thus, available tools do not allow reliable estimation of damage to economic activities in Italy. Indeed, even the MOVIDA project (led by the Po
River District Authority in collaboration with several Italian universities and research centres), representing one of the most recent attempts to define a procedure to estimate flood damage in the Italian context, refrained from estimating direct damage to economic activities, limiting the analysis to the estimation of their exposed value (Simonelli et al., 2022).

## 3 Data

The dataset analysed in the present work is composed of 812 observed damage records to individual firms, collected after five
flood events in Italy:

- The flood occurred in the town of Lodi (Northern Italy), in November 2002, due to the overflow of the Adda river (Molinari et al. 2019, Molinari et al. 2020).
- The extensive flood affecting the Sardegna Region (Insular Italy) in November 2013. Collected data refer to the city of Olbia.
- The flood occurred in the province of Modena (Northern Italy), in January 2014 (Carisi et al. 2018), caused by an embankment breach along the Secchia river. Data refer to three municipalities: Bastiglia, Bomporto, and Modena.





- The flood of the Enza river in the Emilia Romagna Region (Northern Italy) in December 2017; affecting the town of Lentigione (Regione Emilia Romagna, 2018).

- The flood of the Misa river, in September 2023, in the Marche region (Central Italy). Data on affected economic activities regards the municipalities of Ostra, Senigallia and Trecastelli.

Data from the different case studies were merged in a unique dataset to obtain a larger data sample that could be representative of the entire Italy's geographical diversity. Indeed, all case studies refer to riverine flood events occurred in urban context. The dataset was then partitioned into two groups (see Table 2) according to the available information on observed damage and related explanatory variables.

Indeed, while all the data originate from declarations filled in by entrepreneurs in the aftermath of the floods to claim compensation of damages from the government, the level of detail of the information collected for the various case studies is different for two main reasons: (i) case studies refer to different years and regions, each with distinct regulations and standards for data collection, and (ii) the collected data were previously pre-processed by different authorities, ranging from local to regional, responsible for damage compensation. More specifically, a first group comprises 325 observations providing information on the total declared damage, its breakdown by components (structure, equipment, and stock), the estimated water depth at the premises location, the surface occupied by the economic activity, and the NACE code of the economic activity. For a second group, consisting of 487 observations, information is limited to the total amount of the damage, the damage to the structure, the surface, and the NACE code.

**Table 2**. Number of observations per type of available information in the analysed dataset.

| | Damage | | | | Water depth | Activity type | Surface |
|---|---|---|---|---|---|---|---|
| | Total | Structure | Equipment | Stock | | | |
| Group 1 | 325 | 212 | 221 | 271 | 325 | 325 | 325 |
| Group 2 | 487 | 449 | - | - | - | 487 | 487 |
| Entire dataset | 812 | 661 | 221 | 271 | 325 | 812 | 325 |

For Group 1, data referring to hazard (i.e., water depth) come from the hydraulic simulation of the flood events, providing the perimeter of the flooded area and the spatial distribution of the water depth, for each case study (Amadio et al. 2019; Scorzini et al. 2018; Carisi et al. 2018; Gatti 2016). Thanks to the knowledge of the address of affected businesses, it was possible to estimate the water depth at the premises location.

Damage data refer to the full replacement or repair cost of damaged assets. This value is commonly used for detailed micro-scale analysis when depreciated values of assets (i.e., the value at the time the damage occurred) are not available (Allaire, 2018; Balbi et al., 2013). Depending on the case study, damage values are available for the different components of the economic activity. However, for the scope of the analysis, all the data were aggregated into the total damage and, when possible, into three main damage components: structure, equipment, and stock. "Structure" identifies the building in which the economic activity is located with the internal systems necessary for its functioning (e.g., electrical or heating system);





"equipment" refers to machinery, furniture, vehicles, and tools necessary for the functioning of the business; "stock" refers to raw materials, semi-finished and finished products. The choice of the three components is coherent with previous studies (Grelot and Richert, 2020; Kreibich et al., 2010; Schoppa et al., 2020; CGGD, 2018; Booysen et al., 1999). Indeed, it is expected that flood damage mechanisms are different among them, thus requiring different methods to assess damage. All the data were converted to 2022 prices using the consumer price index for comparability reasons.

The NACE code is available at the first level. Although the use of the NACE nomenclature to represent the vulnerability of an activity has been questioned (Molinari et al. 2019; Kreibich et al., 2010), we decided to maintain this information to exploit statistical data on the business sector elaborated by ISTAT (that are organised through the NACE code) and to obtain a description of the economic activities that is shared at the European level (Bremond et al.2020; Paprotny et al.2020). The following categories (C, G, I, J, K, M, N, Q, S) were included in the analysis (see Table 3). In detail, the categories G (wholesale

and retail trade), C (manufacturing), I (accommodation and food service activities), Q (healthcare and social care) and S (other services activities) were considered and renamed for simplicity "Commercial", "Manufacturing", "Restaurant", "Healthcare" and "Services", respectively. The categories J (information and communication), K (financial and insurance activities), M (professional, scientific, and technical activities) and N (administrative and support service activities) were aggregated in a unique category named "Office", since it was assumed the office configuration as prevalent for all the activities referring to

these NACE codes. Some categories have been neglected as they are usually considered separate sectors of investigation in flood risk analysis (Merz et al., 2010) with respect to business activities. This is the case of category A (agricultural, forestry and fishing); categories D (electricity, gas, steam, and air conditioning supply); E (water supply, sewerage, waste management and remediation activities); H (transporting and storage) that refer to infrastructures; O (public administration and defence, compulsory social security) and P (Education) that were considered part of the public sector and strategic infrastructures. Other

categories were instead neglected as they present specific peculiarities that impede a proper comparison; this is the case of category F (Construction) whose elements cannot be associated with a "typical" economic activity configuration of one/some premises with contents, and the category of real estate activities (L) because the damage and the exposed values could refer to the several properties owned or managed by the business, that could be spread in the territory, beyond the flooded areas. The NACE categories B (extraction of minerals from quarries and mines) and R (arts and sports activities) were not investigated

because of the limited number of available data.

Table 3 shows that 46% of the flooded activities were Commercial, 17% Manufacturing, 14% Restaurant, 14% Offices, 3% Healthcare and 6% Services.

The surface occupied by the economic activity is not always available in the original data. In case of its absence, it was approximated with the footprint area of the building in which the activity is located, obtained by Topographic Databases.






**Table 3.** Number of economic activities per NACE code in the entire dataset.

| Category | NACE code | | Number | Percentage |
|---|---|---|---|---|
| Manufacturing | C | Manufacturing | 142 | (17%) |
| Commercial | G | Wholesale and retail trade | 371 | (46%) |
| Restaurant | I | Accommodation and food service | 114 | (14%) |
| | J | Information and communication | 13 | |
| Office | K | Financial and insurance | 10 | (14%) |
| | M | Professional, scientific, and technical | 67 | |
| | N | Administrative and support services | 23 | |
| Healthcare | Q | Healthcare and social work | 22 | (3%) |
| Services | S | Other services | 50 | (6%) |
| | | Total | 812 | |

## 4 Method

The conceptual model at the basis of the analysis is reported in Eq.1, where damage (d) is expressed as a function of three
variables: activity type (*N*), activity surface (*A*) and water depth (*WD*). Such variables are identified in the literature as the
most explicative of damage (Merz et al., 2010; Paprotny et al., 2020; Penning-Rowsell, 2005; Schoppa et al., 2020).

$$d = f(N, A, WD) \tag{1}$$

First, a correlation analysis was implemented to explore the relationship between the damage and the explanatory variables,
as well as to detect potential multicollinearity among them. Then, a damage model was derived for each economic category
included in the analysis. The analysis was carried out firstly for the total damage. Subsequently, we delved into an examination
of damage by components, using the same approach used for the total damage. However, only preliminary insights were
obtained in this case because of the low size of the available data sample.

The damage value used in all the analyses is the unitary damage, obtained dividing the damage (*D*) by the surface of the
economic activity Eq. (2), as better discussed in the following section:

$$d(€/m^2) = \frac{D(€)}{A(m^2)} \tag{2}$$

## 5 Results: total damage

### 5.1. Descriptive statistics

Figure 1 shows the distribution of the unitary total damage, the surface, and the water depth in the observed data. The unitary
total damage (Fig.1a) varies from a minimum value of approximately 1 €/m² to a maximum value of approximately 10,000





€/m², with the first quartile, $Q_1$, equal to 26 €/m², the third, $Q_3$, equal to 277 €/m², the median being 91 €/m² and the average 280 €/m². Figures 1b and 1c shows statistics for the explanatory variables: the firm surface varies from approximately 5 m² to 70,000 m², $Q_1$ is 147 m² and $Q_3$ is 1055 m², the median is 392 m² and the average is 1100 m²; the water depth varies from around 0.05 m to around 3.00 m, with the first quintile, $Q_1$, equal to 0.40 m and, the third quintile, $Q_3$, equal to 1.00 m. the median and the average are approximately 0.70 m. As expected, values for the surface area and the total damage spans over

orders of magnitude, while water depths have relatively little variability, with 50% of the data being concentrated in a small range (0.4-1 m).

Table 4 reports the average values of the unitary total damage and the business surface by economic category, while their distribution is reported in Fig. 2 and 3. Detectable differences suggest that both the surface and the economic category may be relevant explanatory variables, to be included in a damage model.


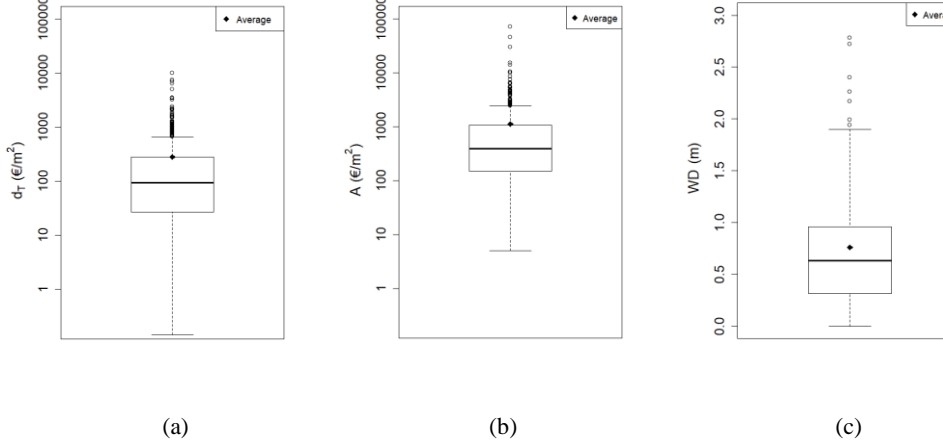

(a)                    (b)                    (c)

**Figure 1: (a)** Distribution of the unitary total damage ($d_T$), **(b)** the surface ($A$) and the **(c)** water depth (*WD*) in observed data. *WD* values refer to Group 1 only. The rhombus represents the average value. The central rectangular box represents the interquartile range (IQR), which

spans from the first quartile ($Q_1$) to the third quartile ($Q_3$), including 50% of the data. The black line inside the box represents the median ($Q_2$). The "whiskers" (i.e. the two lines extending above and below the box) indicates the minimum and maximum values within the range equal to 1.5 times the IQR. Any data points beyond the whiskers are plotted individually.



**Table 4**. Average data values for the unitary total damage ($d_T$), the total damage ($D_T$), and the surface of economic activities ($A$) by economic category, for the entire dataset (group 1 and 2).

| Economic sectors | Average values | | |
|---|---|---|---|
| | $d_T$ (€/m$_2$) | $D_T$ (€) | $A$ (m$^2$) |
| Manufacturing (C) | 250 | 191,000 | 1,500 |
| Commercial (G) | 250 | 100,700 | 1,100 |
| Restaurant (I) | 390 | 121,200 | 870 |
| Healthcare (Q) | 920 | 195,200 | 310 |
| Services (S) | 130 | 33,600 | 640 |
| Offices (J, K, M, N) | 220 | 30,900 | 1,200 |
| Total | 280 | 108,200 | 1,100 |
| Total standard dev. | 670 | 320,100 | 3400 |


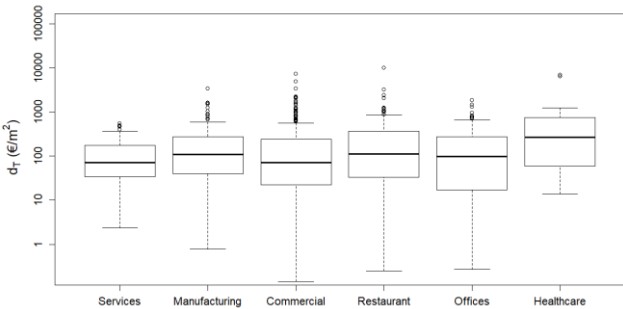

**Figure 2:** Boxplots of the unitary total damage by economic category for the entire dataset (ln-scale).

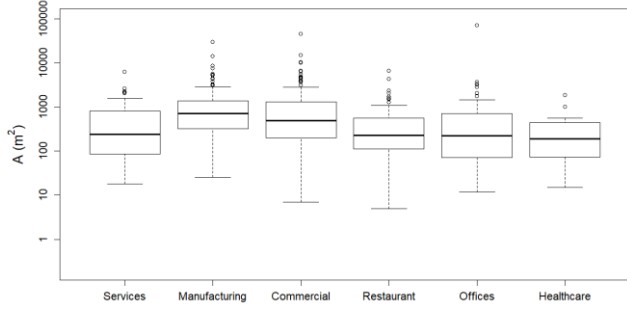

**Figure 3:** Boxplot of the Surface ($A$) of the economic activities by economic category for the entire dataset (ln-scale).



## 5.2 Correlation analysis

The entire dataset was used to test the correlation between the unitary damage and the business surface (Fig. 4a), and only group 1 for the correlation between the unitary damage and the water depth (Fig 4b). Figure 4 plots, for each correlation, outcomes for a Loess (Locally Weighted Scatterplot Smoothing) regression, together with the confidence interval of the curve estimate. Notice that Loess is a non-parametric regression that fits a smooth curve to the data by considering the local density of the data points, thus suggesting possible non-linear trends between the variables.

Figure 4a shows a negative correlation between the unitary damage and the firm surface, indicating that damage increases less than proportionally with respect to the surface. Figure 4b shows a weak positive correlation between the damage and the water depth. The Loess curve is useful to identify a threshold of the positive correlation, with no effect on damages for water depth values smaller than 0.2 m. The damage-surface and damage-water depth correlations are, however, influenced by the existence of a negative correlation between the surface and the water depth (Fig.4 c); therefore, although the positive (although here weak) correlation between water depth and damage is physically sound, its statistical evidence may be masked by the water depth-surface collinearity in the dataset (which has no evident physical meaning).

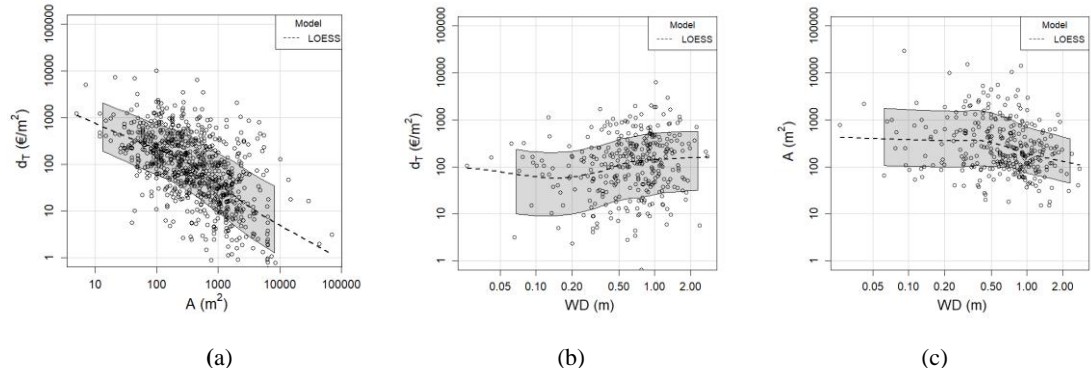

|     (a)     |     (b)     |     (c)     |

**Figure 4: (a)** Unitary damage ($d_T$) vs firm surface ($A$), entire dataset. **(b)** Unitary damage ($d_T$) vs water depth ($WD$), group 1. **(c)** Firm surface ($A$) vs water depth ($WD$), group 1. The dashed curve represents the results of the Loess regression. The grey region indicates the confidence interval of the curve estimate.

## 5.3 Forecast model

The following function has been assumed to describe the relationship between the unitary damage and the explanatory variables:

$$d_T = e^{\beta_0} * A^{\beta_1} * WD^{\beta_2} * e^{\beta_3 D_C} * e^{\beta_4 D_I} * e^{\beta_5 D_Q} * e^{\beta_6 D_{Off}} * e^{\beta_7 D_S} \tag{2}$$

where:



- $D_C$, is the dummy variable for the manufacture category.
- $D_I$, is the dummy variable for the restaurant category.
- $D_{off}$, is the dummy variable for the offices category.
- $D_Q$, is the dummy variable for the healthcare category.
- $D_S$, is the dummy variable for the services category.

The decision to use a product function for damage estimation was primarily driven by the need to account for a relative
independence of the firm's size and/or water depth on the damage. This implies that, according to equation (3), any variation
in business size or water depth will lead to a proportional variation in damage. The decision to use dummy variables to account
for the variation in damage across economic categories is driven by the limited size of the dataset, which is insufficient to
create a separate econometric model for each category. The model accordingly assumes the impact of surface and water depth
on damage as uniform across all economic categories.

The model assumes 'commercial' as the reference category; therefore, no dummy variable is introduced for this category. If
data pertain to another sector (say, manufacture), the corresponding dummy variable is equal to unity, while all the others are
zero; consequently, $e^{\beta_3 D_C}$ amplifies ($\beta_3 > 0$) or diminishes ($\beta_3 < 0$) the reference constant $e^{\beta_0}$, while all the other groups $e^{\beta_x D_x}$
are neutral in the regression as they assume unity values.

A logarithmic transform turns equation (3) into equation (4), to be estimated by Ordinary Least Squares (OLS) after adding *u,*
the random error term, representing the unobserved factors that affect the damage but are not included in the model.

$$ln(d_T) = \beta_0 + \beta_1 \, ln(A) + \beta_2 \, ln(WD) + \beta_3 D_C + \beta_4 D_I + \beta_5 D_{off} + \beta_6 D_Q + \beta_7 D_s + u \qquad (3)$$

Log transformations of variables also have some statistical advantages in the case of skewed distributions as those in Fig. 1:
by taking the natural logarithm of the values, the distribution becomes more symmetrical, allowing a more reliable calculation
of some statistical measures that assume the data are approximately normally distributed. Using log transformed variables the
variability of both dependent and independent variables is reduced, and it permits to decrease the susceptibility of Ordinary
Least Squares (OLS) estimates to extreme (outlying) values; thus, there is no need to eliminate them.

The forecast model was initially derived using only the data from Group 1, the dataset for which all explanatory variables are
available. The results are provided in the first column (1) of Table 5, indicating the lack of statistical significance for water
depth, likely because the variable does not vary enough to significantly influence the damage caused. Subsequently, water
depth was dropped as explanatory variable, and the analysis was extended to the entire dataset (Group 1 + 2) to enhance model
robustness. For the same reason, the K-fold cross-validation technique was implemented (Hasanzadeh Nafari et al., 2016). The
dataset was divided into K subsets (folds) of approximately equal size, with K set to 10. The model was then trained and
evaluated K times, each time using one of the folds as the test set and the remaining (K-1) folds as the training set. The second
column (2) of Table 5 presents the results of model training and the results of model validation.





Results reveal a negative and statistically significant coefficient for the business's surface ($p < 0.01$), confirming the correlation observed in the previous section. In the model with Group 1, differences across categories are evident only for damage to manufacturing, which is higher than the others. By adding observations (Groups 1+2), it becomes apparent that damage to offices ($D_{off}$) is also statistically significant. Specifically, the manufacturing ($D_c$) category exhibits higher damage (by 88%), while offices have lower damage (by 67%) compared to the commercial category. Concerning the surface coefficient, for a

10% increase in the size of the firm, the unitary damage decreases, on average, by around 8%.

The goodness of fit, expressed by the adjusted R² in Table 5, is 0.37, indicating that the model can explain 37% of the total variance in damage. The coefficient of variation (CV) indicates that, on average, individual point errors are approximately two and a half times the mean of the response variable.

According to the analysis, the forecast model for the damage can be expressed as:

$$d_T = 8022 * A^{-0.80} * 1.88 D_c * 0.60 D_{off} \tag{4}$$

**Table 5.** Results of the log-log regression with OLS for the unitary total damage; standard errors in brackets. The magnitude of the statistical significance of the explicative variables is expressed by the number of stars (*). A low p-value (***), indicates greater evidence against the null hypothesis and thus greater statistical significance: $*p<0.1; **p<0.05; ***p<0.01$.

| | (1) Group 1 | (2) Group 1+2 |
|---|---|---|
| Variables | ln ($d_T$) | ln ($d_T$) |
| Constant | 8.43*** | 8.99*** |
| | (0.37) | (0.22) |
| ln (A) | -0.68*** | -0.78*** |
| | (0.06) | (0.03) |
| ln (WD) | 0.12 | - |
| | (0.09) | |
| $D_C$ | 0.58** | 0.63*** |
| | (0.20) | (0.13) |
| $D_I$ | -0.02 | |
| | (0.24) | |
| $D_{off}$ | -0.24 | -0.51*** |
| | (0.22) | (0.14) |
| $D_Q$ | 0.56 | - |
| | (0.37) | |
| $D_S$ | -0.33 | - |
| | (0.27) | |
| R² adj. | 0.30 | 0.37 |
| F-statistics | 20.09 | 157 |
| N. of obs. | 325 | 812 |
| Average RMSE | | 661 |
| CV | | 2.4 |



# 6 Results: damage to components

This paragraph replicates the analysis made in previous section, albeit with a distinction among the damage components.

## 6.1 Descriptive statistics

The analysed variables are the unitary damage to the building structure ($d_{BS}$), stock ($d_S$) and equipment ($d_E$) in relation to the surface of the activity ($A$) and the water depths ($WD$):

Table 7 presents the average values of unitary damage by component and economic category, while their distribution is

depicted in Fig.5. Across all categories, the highest average damage is observed for equipment, followed by stock and structure. Similar findings were reported by Samantha (2018). It is evident that for the unitary damage to the building structure and equipment, the highest median values are observed for the Healthcare, Offices, and Restaurant categories, which have the lowest surface area (Fig. 3) and are often located in civil buildings. For damage to the stock, the highest median value is associated with the commercial category.

**Table 7.** Average values of the damage by components and by economic categories. Damage to the structure (BS) includes data from the entire dataset; damages for the equipment (E) and stock (S) include only the data of the Group 1.

| Economic sectors | Average values | | | | | |
|---|---|---|---|---|---|---|
| | $d_{BS}$ (€/m²) | $d_S$ (€/m²) | $d_E$ (€/m²) | $D_{BS}$ (€) | $D_S$ (€) | $D_E$ (€) |
| Manufacturing (C) | 70 | 90 | 140 | 63,700 | 73,300 | 83,500 |
| Commercial (G) | 80 | 240 | 140 | 41,700 | 56,100 | 2700 |
| Restaurant (I) | 240 | 50 | 220 | 58,200 | 5,800 | 48,800 |
| Health (Q) | 630 | 90 | 300 | 109,600 | 5,200 | 23,300 |
| Services (S) | 70 | 70 | 80 | 18,400 | 10,300 | 16,200 |
| Offices (J, K, M, N) | 120 | 50 | 190 | 20,700 | 4,700 | 15,700 |
| Total average | 120 | 140 | 161 | 45,500 | 42,800 | 41,200 |

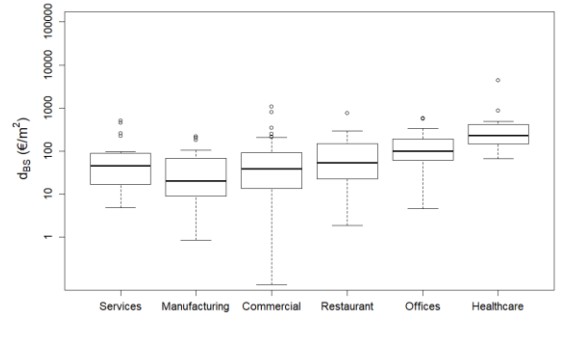

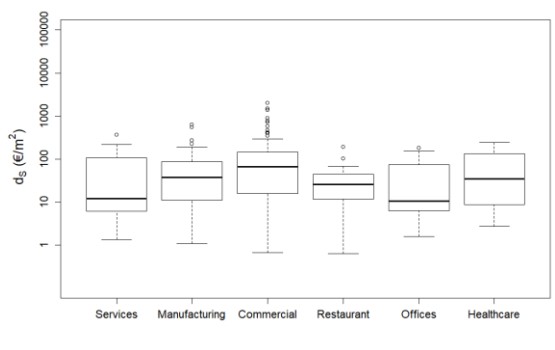

(a)                                                                  (b)





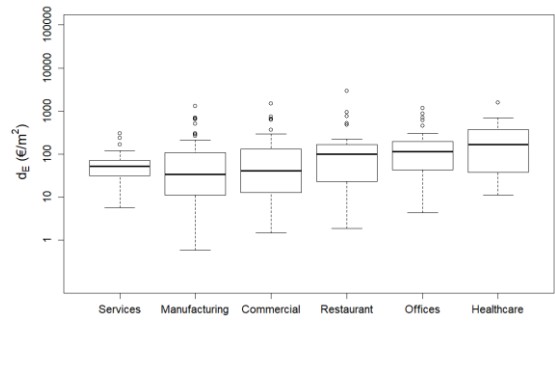


(c)

**Figure 5:** Boxplot of the unitary damage by component and by economic category (ln-scale); **(a)** damage to the structure ($d_{BS}$); **(b)** damage to the stock ($d_S$); **(c)** damage to the equipment ($d_E$).

## 6.2 Correlation analysis

Figures 6 and 7 illustrate the correlation between the unitary damage incurred for various components and the surface area of the firm, along with the water depth level. The correlation with the surface is negative for all the components, confirming the trend observed for the unitary total damage (Fig. 4). In Figure 7, the Loess curve indicates a positive correlation with water depth, with the stock component showing the most pronounced dependence.

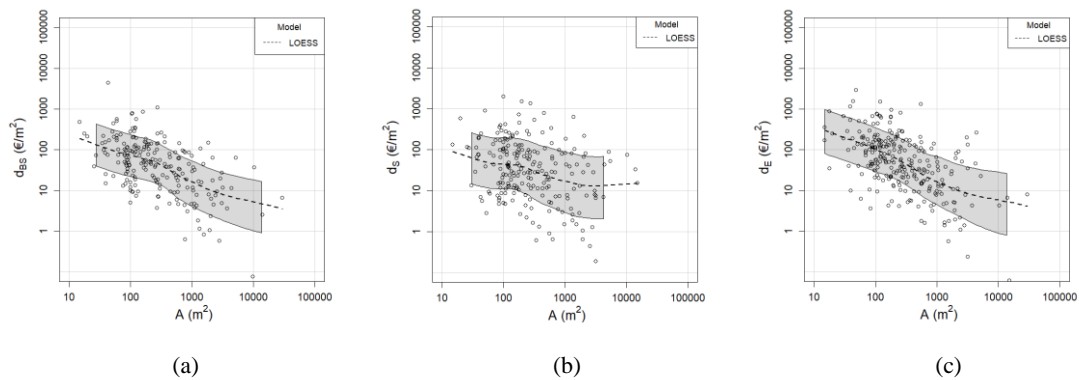

350                     (a)                           (b)                       (c)

**Figure 6**: Scatterplot of the unitary damage vs firm surface. **(a)** Unitary damage to the structure ($d_{BS}$). **(b)** Unitary damage to the stock ($d_S$). **(c)** Unitary damage to the equipment ($d_E$).





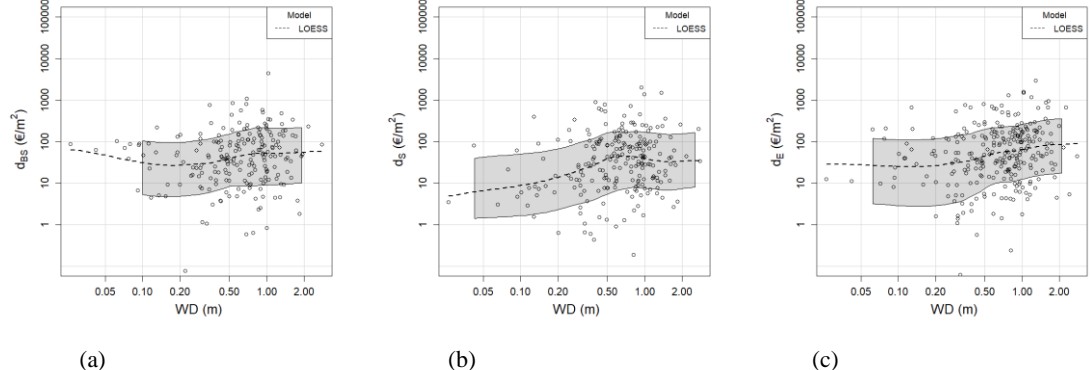

(a)            (b)            (c)

**Figure 7**: Scatterplot of the unitary damage vs the water depth **(a)** Unitary damage to the structure ($d_{BS}$). **(b)** Unitary damage to the stock ($d_S$). **(c)** Unitary damage to the equipment ($d_E$).

### 6.3 Forecast model

The functional relations tested for damage to components is analogous to that for the total damage:

$$d_{BS,S,E}=e^{\beta_0} * A^{\beta_1} * WD^{\beta_2} * e^{\beta_3 D_C} * e^{\beta_4 D_I} * e^{\beta_5 D_Q} * e^{\beta_6 D_{Off}} * e^{\beta_7 D_S} \tag{5}$$

Table 8 shows the results of the linear log-log regression model used to derive the forecast model. The analysis was performed without validation, using the whole dataset due to the limited number of observations. Specifically, columns *(1), (3), (5)* present the results of the initial regression, including all explanatory variables. Subsequently, columns *(2), (4), (6)* present the outcomes of a refined regression, considering only the variables with statistically significant coefficients.

The results align with the correlation analysis. The coefficients for the surface are negative and significant for all damage components, signifying that, as for the total damage, as the size of the business increases, the unit damage decreases by an amount that depends on the initial value of *A*. The coefficient of water depth is positive and significant only for stock damage, indicating that as the water depth increases, the unit damage increases by an amount that depends on the initial value of *WD*. In terms of differences between damage in different categories, statistically significant variations are observed. For damage to structure, only the Healthcare category shows significantly higher average damage compared to the Commercial category. In the case of equipment damage, the Manufacturing category stands out with statistically higher damage than the Commercial category. Regarding stock damage, all categories, except Manufacturing, exhibit lower average damage than the Commercial category.

The goodness of the fit expressed by the adjusted $R^2$ varies from 0.26 to 0.36, meaning that the model explains from 26% to 36% of the damage to the different components.

According to the analysis, the forecast model for the unitary damage to the components can be expressed as:

$$d_{BS}=1118 * A^{(-0.61)} * 4.18 D_Q \tag{11}$$





$$d_S = 1339 * A^{(-0.55)} * WD^{(0.35)} * 0{,}29D_I * 0.16D_{off} * 0.23D_Q * 0.26(D_S) \tag{12}$$

$$d_E = 4491 * A^{(-0.84)} * 1.88D_C \tag{13}$$


**Table 8.** Result of the log-log regression model for each component of the damage: unitary damage to the structure ($d_{BS}$), unitary damage to the stock ($d_S$), unitary damage to the equipment ($d_E$). Standard errors in brackets. The magnitude of the statistical significance of the explicative variables is expressed by the number of stars (*). A low p-value (***), indicates greater evidence against the null hypothesis and thus greater statistical significance: $*p<0.1$; $**p<0.05$; $***p<0.01$.

| | (1) | (2) | (3) | (4) | (5) | (6) |
|---|---|---|---|---|---|---|
| Variables | ln ($d_{BS}$) | ln ($d_{BS}$) | ln ($d_S$) | ln ($d_S$) | ln ($d_E$) | ln ($d_E$) |
| Constant | 6.98*** | 7.02*** | 7.21 *** | 7.20*** | 8.17 *** | 8.41*** |
| | (0.44) | (0.36) | (0.49) | (0.49) | (0.44) | (0.38) |
| ln (A) | -0.63 *** | -0.61*** | -0.55 *** | -0.55*** | -0.81 *** | -0.84*** |
| | (0.07) | (0.06) | (0.08) | (0.08) | (-0.81) | (0.07) |
| ln (WD) | 0.01 | - | 0.35 ** | 0.35** | 0.15 | - |
| | (0.11) | | (0.13) | (0.13) | (0.11) | |
| $D_C$ | 0.31 | - | 0.06 | - | 0.75*** | 0.63** |
| | (0.23) | | (0.26) | | (0.22) | (0.21) |
| $D_I$ | 0.12 | - | -1.28*** | -1.30*** | 0.43 | - |
| | (0.28) | | (0.30) | (0.29) | (0.27) | |
| $D_{off}$ | 0.42 | - | -1.84 *** | -1.86*** | 0.17 | - |
| | (0.28) | | (0.34) | (0.33) | (0.30) | |
| $D_Q$ | 1.55 *** | 1.43* | -1.49* | -1.51** | 0.50 | - |
| | (1.55) | (0.39) | (0.58) | (0.57) | (0.42) | |
| $D_S$ | -0.17 | - | -1.34 *** | -1.37*** | -0.26 | - |
| | (-0.17) | | (0.40) | (0.38) | (0.32) | |
| $R^2$ adj. | 0.36 | 0.36 | 0.25 | 0.26 | 0.37 | 0.36 |
| F-statistics | 18 | 69 | 12 | 14 | 23 | 77 |
| N. of obs. | 212 | 212 | 221 | 221 | 271 | 271 |

**7 Performance compared to foreign models**

We conducted a comparison between the estimations provided by our model and those of existing models in Europe to determine added value with respect to the inadvisable implementation of foreign models. Specifically, the comparison was made with models discussed in section 2, namely FLEMO-cs, the model included in the Multicoloured Manual (referred to as MCM hereafter), and the French model (https://www.ecologie.gouv.fr/politiques-publiques/levaluation-economique-projets-
gestion-risques-naturels) This comparison was conducted using data only from group 1, as foreign models typically require



input values for water depth at the premises' location, which are available only for this dataset. Additionally, with the exception of the French model, the foreign models are relative, necessitating a preliminary estimation of the exposed value to achieve damage estimations that are comparable among all the models under consideration. To accomplish this task, we implemented the method based on the net capital stock proposed by Molinari et al. (2016) and further implemented in the MOVIDA project.

This approach has two main limitations: (i) the uncertainty of estimated exposed values, and (ii) the possibility to compare only damage to structure and equipment, as net capital stock values are supplied by ISTAT only at the national level and without reference to stock; however, a distinction is made among the various NACE categories. Nonetheless, it was not possible to compare all the models for all the categories included in our model, as they are not always present in foreign models. Table 9 provides a synthesis of the main characteristics of compared models. With specific reference to the French

model, even though it provides a direct estimation of damage and includes all categories, the comparison was possible only for the damage to the structure component, as information on the number of employees required for the estimation of the damage to equipment is not present in our dataset. The comparison with available Italian models was not conducted for the following reasons. As discussed in section 2, the model of Arrighi et al. is closely tied to the exposure and vulnerability parameters of the city of Florence. It relies on local real estate values specific to the city, making it difficult to compare with

national net capital stock values.

**Table 9.** Main characteristics of models used for comparison.

| Model | Damage type | Economic category | Measure of damage |
|---|---|---|---|
| *FLEMO-cs* | Structure | Public and private services | % of the exposed value |
| | Equipment | Industry | |
| | Stock | Commercial | |
| *France* | Structure | All categories | €/m² (Structure) |
| | Equipment | | €/employee (equipment) |
| *MCM* | Structure | Commercial | % of the exposed value |
| | Equipment | | |
| | Stock | | |
| *Ballocci et al. (this study)* | Structure | Commercial | €/m² |
| | Equipment | Manufacturing | |
| | Stock | Restaurant | |
| | Total | Services | |
| | | Healthcare | |
| | | Offices | |

On the other hand, the model of Molinari et al. 2020 implements a foreign meso-scale model, rendering the comparison with our micro-scale model nonsensical a priori.





Table 10 and 11 display the results of the comparison in terms of the RMSE calculated with reference to the logarithm of the
unitary damage, considering the nature of our model that minimizes relative errors rather than absolute ones. The findings
underscore that our model exhibits the best performance, thereby enhancing the reliability of damage estimation for the Italian
context. This result was expected, since the model was built on this data set. However, this comparison gives a measure of the
reduction in uncertainty (error) that we would have had in the case of directly applying the other models to the Italian context.

**Table 10.** RMSE of the logarithm of the unitary damage to the structure (€/m²). "n.a." indicates that the model does not provide an estimation
for the corresponding category.

| | Model | | | |
|---|---|---|---|---|
| Category | France | FLEMO-cs | MCM | Ballocci et al. |
| Manufacturing | 1,53 | 1,46 | 1,17 | 1,14 |
| Commercial | 1,67 | 1,86 | 1,65 | 1,42 |
| Restaurant | 1,68 | n.a. | 1,52 | 1,26 |
| Offices | 1,35 | n.a. | 1,65 | 0,96 |
| Healthcare | 2,05 | n.a | n.a. | 1,07 |
| Services | 1,33 | n.a | n.a. | 0,95 |

**Table 11.** RMSE of the logarithm of the unitary damage to equipment (€/m²). "n.a." indicates that the model does not provide an estimation
for the corresponding category, or that our data do not allow the model implementation (i.e., French model). Only categories for which at
least one foreign model is available are reported.

| | Model | | | |
|---|---|---|---|---|
| Category | France | FLEMO-cs | MCM | Ballocci et al. |
| Manufacturing | n.a. | 5,45 | 1,85 | 1,51 |
| Commercial | n.a. | 1,69 | 1,85 | 1,36 |
| Restaurant | n.a. | n.a. | 1,52 | 1,28 |
| Offices | n.a. | n.a. | 1,43 | 0,72 |

**8 Discussion**

The adopted modelling approach proved to be valuable for both comprehending the role of each explanatory variable in the
definition of the damage to different business components and detecting differences among damages occurring in different
business categories.

Results indicate a significant impact of the firm's surface on the extent of damage, albeit with varying degrees for different
damage components (structure, equipment, stock, and total). Notably, water depth exhibits a significant influence only on
stock damage. It is worth noting that the limited variation in water depth within the data sample (with 50% of values ranging
between 0.40 m and 1 m) may inhibit to establish a statistically significant relationship with the overall damage.



The study identifies Healthcare, Commercial, and Manufacturing as the most vulnerable categories for building structure, stock, and equipment, respectively. Several factors may contribute to the higher damage in the manufacturing category. One possibility is the presence of more expensive and specialized equipment, which could be more susceptible to flooding compared to equipment in other categories. Additionally, the manufacturing and commercial categories may handle a greater quantity of goods, and the nature of these goods (such as food, electronics, or chemicals) could make them more sensitive to water damage, resulting in higher damage to stocks. As for the elevated damage to the structure in the Healthcare category, this could be attributed to the typically civil nature of the buildings where they are located, generally more vulnerable than industrial sheds. Overall, the observed variability underscores the importance of tailored risk management approaches for different business categories.

However, despite an improvement in damage estimation reliability compared to the implementation of foreign models in the Italian context, the model's predictive capability is still considered unsatisfactory. The goodness of fit to the observed data indicates that the model can only account for around 37% of the variance in total unitary damage and between 26% and 36% of the variance in individual components. Additionally, the root mean square error (RMSE) for total unitary damage is approximately double the average unitary damage. We attribute these limitations mainly to constraints within the available dataset. It is evident that crucial explanatory variables are missing, or in other words, the variables incorporated in the model might not be capturing all the relevant damage mechanisms. Moreover, as explained in section 3, available data are characterized by varying levels of reliability, contingent upon the specific case study to which they refer, and thus, on the methods employed for their collection and evaluation. Another potential concern is the presence of a "selection bias" arising from the counterintuitive negative relationship between the building's surface area and water depth. This bias might be elucidated by an underrepresentation of small enterprises with shallow water depths in the dataset. Specifically, the dataset predominantly includes large enterprises (i.e., exceeding 80 m$^2$) with water depths lower than the average value, resulting in high absolute damage (Fig. 4, b). Conversely, small enterprises (below 80 m$^2$) with water depths higher than the average value exhibit lower absolute damage (see Appendix Fig. 1). This suggests the possibility of missing values associated with small economic activities with low water depth. This absence of information could impact both correlations depicted in Fig. 4a and 4b. As a matter of fact, it is likely that larger businesses, possessing more resources and personnel, are more inclined to report damages compared to smaller businesses. The increased likelihood of larger businesses engaging in the reporting process could stem from their ability to allocate resources to such endeavours and navigate through bureaucratic procedures. In contrast, smaller businesses, constrained by limited resources and personnel, may find the process more burdensome, especially when there are low expectations of receiving compensation, leading to underrepresentation in the reported data. The knowledge about flooded companies that suffered damage but did not declare it could support the testing of the selection bias by means of the "Heckman model"(Heckman, 1979). The latter would be able to test if the likelihood of an observation to be within the sample is related to the characteristics of the business and to correct the coefficients estimation for missing data.



Overall, our results point to the need of disposing of more complete and numerous data, specifically collected for the objective of damage modelling (Ballio et al., 2018; Pogliani et al., 2021). This could also lead to the possibility of performing different econometric analysis for different business categories.

On the other hand, the obtained results were predictable. It is well known that micro-scale damage models exhibit significant dispersion with respect to observed data. This phenomenon applies even to residential buildings, which are characterized by
less heterogeneity than economic activities and for which more reliable data generally exist (Molinari et al., 2020).

In terms of feasibility, our model overcomes the challenge of reliably estimating the value of exposed assets in Italy, as discussed in the previous section. On the contrary, as it provides estimates of damage in absolute terms, it also requires updating the estimated damage value to the current currency, posing a further obstacle to its exportability to foreign countries. The model can be applied to estimate damage to economic activities across the entire Italian territory in cases of riverine floods. It
is derived from data collected from various regions of the country and relies on input data commonly available throughout Italy, such as the surface area of the activity and the NACE code. Importantly, its independence from the water depth enables damage estimation in the early aftermath of a flood, when only the perimeter of the flooded area is typically known, as well as in areas lacking 2D hydraulic modelling. However, given the uncertainty inherent in the model, we recommend its implementation for comparative purposes, such as comparing risk levels across different areas, rather than for reliable
estimation of risk in a specific region.

**9 Conclusions**

The analysis conducted in this study provides significant new insights into the direct damage caused by floods to economic activities in Italy. By employing an econometric approach to empirical damage data, the study enhances our understanding of the impact of explanatory variables on damage to different business components and identifies variations in damages across
different business categories, enabling the adoption of tailored risk management approaches. Furthermore, it presents a forecast model that outperforms foreign models and can be utilized for comparative purposes across the entire Italian territory to estimate flood risk/damage, both in post-event scenarios and in the peace time, where riverine floods are of concern. Overall, this study underscores the necessity of comprehensive and numerous empirical data specifically collected for damage modelling purposes to enhance our understanding of damage mechanisms and improve our capability for damage prediction.






**Appendix A**

**1 Selection bias**

In Figure A1 is represented the average total damage by class of water depth with the number of enterprises for small and large
enterprises. The aim of this figure is to show that in our sample there is a prevalence of large enterprises, with a size larger
than 80 m², with shallow water depth, lower than 0.7 m, which have high absolute damage (€) and small enterprises with high
water depths (higher than 0.7 m) which have lower absolute damage. This leads to the assumption that there could be missing
values related to the small economic activity with low water depth. These possible existence of missing values could lead to a
bias in the econometric model that could be tested with the *"Heckman model"*, if we had data on flooded companies that
suffered damage but did not declare it by filling the form. The Heckman model would be able to test if the likelihood of an
observation to be within the sample could be related to the characteristics of the business.

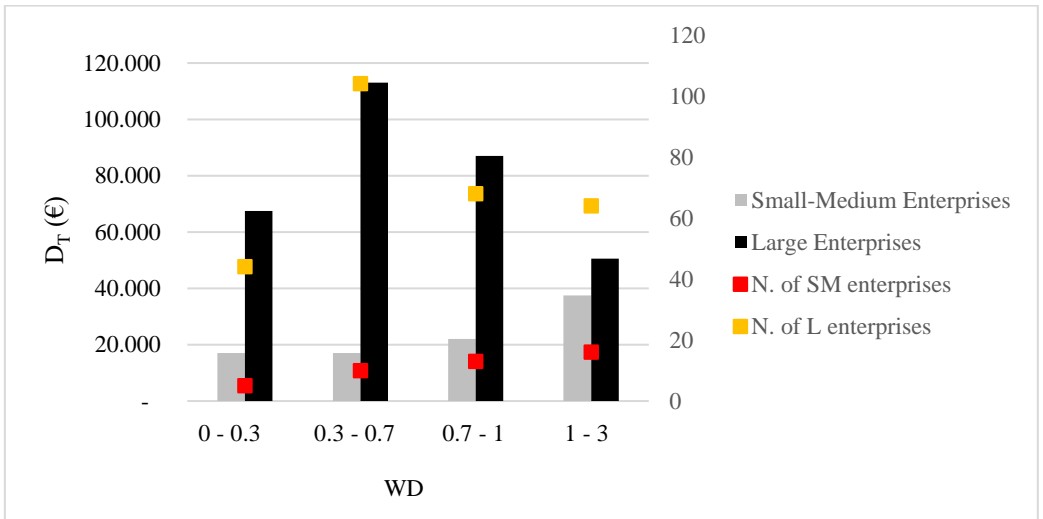

**Figure A1**: Average total damage ($D_T$) by class of water depth (0.0 - 0.3, 0.3 - 0.7,0.7 - 1,1 - 3 m) and by firm's dimension (small-medium
with the surface <80 m² and larger with the surface >80m²).

In fact, the choice by businesses to declare the existence of damage could depend on their own characteristics. Larger
businesses with more employees and turnover may have a higher likelihood of reporting damage than smaller businesses.
Conversely, small businesses with few staff may have fewer resources to employ in damage reporting especially if they
suffered a small damage and the expectation of being compensated may discourage them due to the large bureaucracy. Using
the Heckman model helps address this issue and correct the estimation for missing data related to small businesses with low
water depths.






## 2 OLS assumption validation total damage

Table A1 presents the results validating the OLS assumptions of the total damage model. The Shapiro-Wilk test suggests that the residuals follow an approximately normal distribution (p-value > 0.05). Heteroscedasticity is evident in the residuals, as indicated by the significantly low p-value of the Breusch-Pagan test. However, there is no significant evidence of
autocorrelation in the residuals, as indicated by the high p-value of the Durbin-Watson test.

**Table A1.** Analysis of residuals of OLS specification.

| Test type | $d_T$ |
|---|---|
| Shapiro-Wilk normality test | W= 0.996, p-value = 0.198 |
| Studentized Breusch-Pagan test | BP = 19.3, p-value = 0.0002 |
| Durbin-Watson test | DW = 2.18, p-value = 0.983 |

## 3 OLS assumption validation damage by components

Table A2 presents the results validating the OLS assumptions of the damage model by components. For structure damage, the
analysis indicates potential issues with slight heteroscedasticity and autocorrelation in the model residuals. Concerning stock damage, the residuals do not show significant autocorrelation and may exhibit some evidence of heteroscedasticity, although the latter is not highly significant. For equipment damage, the residuals do not show significant autocorrelation, but there is substantial evidence of heteroscedasticity.

**Table A2.** Analysis of residuals of OLS specification.

| Test type | $d_{BS}$ | $d_S$ | $d_E$ |
|---|---|---|---|
| Shapiro-Wilk normality test | W = 0.9966, p-v. = 0.0794 | W = 0.994, p-v. = 0.5281 | W = 0.99491, p-v. = 0.5085 |
| Studentized Breusch-Pagan test | DW = 1.7714, p-v. = 0.0392 | DW = 1.9621, p-v. = 0.2784 | BP = 10.897, p-v. = 0.0043 |
| Durbin-Watson test | BP = 8.0136, p-v.= 0.0182 | BP=11.763, p-v. = 0.0675 | DW = 1.9828, p-v. = 0.4143 |





**Code availability:** upon request.

**Author contribution**:

Marta Ballocci: data curation, research conceptualisation, data analysis, results investigation, writing-first draft, writing-
review; Daniela Molinari: data curation, research conceptualization, supervision, results investigation, writing-first draft,
writing-review; Francesco Ballio: research conceptualization, supervision, results investigation, writing-review; Giovanni
Marin:  research conceptualisation, supervision; Marta Galliani, data curation, writing – first draft; Alessio Domeneghetti
data curation, writing review; Giovanni Menduni: data curation, writing-review; Simone Sterlacchini: data curation, writing-
review.

**Competing interests**

The authors declare that they have no conflict of interest.

**Disclaimer**

**Acknowledgements**

Authors wish acknowledge with gratitude Alessi Carisi and Marco Zazzeri for their initial support in data curation



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
