# Peer review of "Econometric Modelling for Estimating Direct Flood Damage to Firms: A Micro-Scale Approach Using Post-Event Records in Italy"

_EGUsphere, 2024_

## Referee Comment (RC1)

**Comments on "Econometric Modelling for Estimating Direct Flood Damage to Firms: A Local-Scale Approach Using Post-Event Records in Italy" submitted by Ballocci et al.**

**1 General comments**

5   This is an empirical study analysing the direct flood damage to building, stock and equipment in the business sector. The dataset covers five flood events in several regions of Italy. Regression models are calibrated, by considering dummy variables for the economic sectors to compensate for the relatively low number of observations (especially when disaggregated per sector and/or per direct damage component). The research question is important and timely. Overall, the methodology is sensible.

10   A challenging aspect of the study is the relatively limited sample size (given the number of cost components to be predicted for several sectors). This is a typical challenge in the field. Consequently, the predictive capacity of the regressions is limited, as correctly acknowledged by the Authors (Line 438: "considered unsatisfactory"); but it provides nonetheless valuable insights into the important features, the relative amounts of damage in different sectors and across cost components. My opinion, is that the paper is suitable for publication in this journal, provided the points below are addressed, but it would

15   become even more valuable if the text can be made more concise, by identifying a limited number of key take-home messages of the study, and rewriting the text more straight-to-the-point to convey these few key messages to the reader. Additional material can be moved to a Supplement.

The paper would also benefit from a bit more in-depth information on how data were collected and processed. Particularly, the step of attributing a single, representative value of water depth to (relatively large) business buildings needs to be

20   discussed with more attention.

**2 Specific comments**

- The Authors must make sure that they consistently adhere to a single term to refer to a given concept. If different words are used interchangeably to denote the same idea, it creates confusion for the reader. Examples are as follows:

25
  - firms, enterprises …: if they all mean the same, choose one word and stick to it;
  - local-scale (title), micro-scale, micro-local (Abstract) …
  - "multiple regression model"

- L52: "The assessment of damage is important to evaluate actions from …" Is the model proposed by the Authors suitable for evaluating a wide range of risk reduction measures. It is important to discuss this point in Section 8, particularly in light of regression models which do not consider any hazard explanatory variable.

- L160-164: exactly how data was collected and how it was processed is key to the analysis. I recommend that a Supplement to the paper is created to provide more details on how data was collected and treated for each flood event.

- L170-174: exactly how water depth was estimated is crucial for the entire analysis. The Authors are asked to describe in detail how the hydrodynamic models were validated (which is their performance, and hence estimate the uncertainty on the water depths), and how water depth attribution to (large) buildings was performed. The latter point refers to how a single representative value of water depth was assigned to a particular building from the gridded results of a 2D model. Considering the highest water depth around the building, or the average, or other statistics can lead to substantially different results, especially for large buildings. How were the buildings incorporated in the 2D simulations (building block approach vs. building hole approach …)?

- I'm particularly sceptical of all the boxplots presented with a log-scale $y$-axis extending up to $10^5$ €/m² (Figure 1 and later). This is motivated by the existence of a few outliers; but I believe that this representation leads to a bias in the visual inspection of the plots. I would recommend keeping one version of the boxplots as they are, and including a second version based on a linear-scale bounded by reasonable values (e.g., up to 5,000 €/m²). One version could remain in the main text, and the other one be moved to a Supplement.

- L265-267: this may be related to the technique used for water depth attribution to buildings from the 2D computations.

- L284-286: this is not clear. Please reformulate and be sure to use the right words. Is "proportional" correct?

- L299: what is meant with "some statistical measures". Be more specific. Which one is relevant here?

- L309 and Table 5: it is unclear how the results for model training and validation are displayed. Importantly, it is also not clear what is presented in Table 5 regarding the 10 folds … Does Table 5 display an average over the 10 folds, or something else? This is key to clarify.

- L310-315: it is unclear whether "damage" or "unitary damage" is considered. The text seems to lack rigor. Percentages are given; but it is not clear to what they refer (average value, median value …).

- L316-318: elaborate a bit more on whether this result is satisfactory or not.

- Table 7 could be moved to a Supplement, and replaced by a bar chart in the main text. The last row of Table 7 is not clear "Total average" …

- Paragraph starting in Line 365: the use of "initial value" is not clear and seems not rigorous.

- L375: this seems incorrect. It is a portion of the variance which is explained. Again, bi rigorous with the use of the words "damage" or "unitary damage".

- It is essential that the performance of each "final" model (Eqs. 11 to 13) is represented by means of a scatter plot (computed vs. observed values), and that the possible trends revealed by these plots (over- vs. under-estimations …) are briefly discussed.
- Lines 402 and following: it is not clear which are the "Italian models". Are there only two such models (Arrighi et al. and Molinari et al., 2020), or also others? Besides, make sure to avoid contradictions in the text: L469 it is stated that the new model applies to the "whole Italian territory". Then, also Florence? In that case, how does it compare with Arrighi et al.? It is probably better to tone down the statement "whole Italian territory" and emphasize the regions for which data were available and were used for calibrating the regressions.
- L410: discussion on relative vs. absolute errors is not clear. Please clarify.
- The main results need to be concisely summarized in the Conclusion.

**3 Technical corrections**

Overall, the text is reasonably well written. Here and there, sentences are odd, and weird formulations need to be corrected.

- L22-23: rephrase to improve clarity
- L23: is "vulnerable" the correct word? Data show that these are the highest values of damage. It is not proven that the reason for the higher damage is a higher vulnerability. The hazard may be higher.
- L60: existent
- Remove multiple occurrences of "in fact" throughout the text.
- Many references cited in the text are missing in the reference list.
- Table 2: bottom left cell should read 812 instead of 325.
- L221: same approach as
- In general, avoid repeating generally known information, such as a description of a boxplot (e.g., in caption of Figure 1).
- There are errors in the numbering of the equations.
- L293: "take unity values"
- L299: remove "caused"
- In figures like Figure 7, the exact meaning of the grey shaded area needs to be explained.
- L387. Reformulate "inadvisable".
- L391: remove "at the premises' location".
- L463. Delete the first sentence ("… results were predictable.").

A thorough review of the text for typos is necessary, as well as for improving English style. The items listed above are just examples of necessary improvements.

---

## Author Comment (AC1)

**Econometric Modelling for Estimating Direct Flood Damage to Firms: A Local-Scale Approach Using Post-Event Records in Italy.**

*Ballocci M., Molinari D., Marin G., Galliani M., Domeneghetti A., Menduni G., Sterlacchini S., and Ballio F.*

**Response to the Reviewers' comments**

We thank the Editors and the Reviewer for their interest in our manuscript and the valuable comments they have provided. According to the, the  manuscript will be revised as follow.

A point-by-point response to the reviewers' comments is provided below..

**Reviewer #1**

**1. General Comments**

**"***My opinion, is that the paper is suitable for publication in this journal, provided the points below are addressed, but it would become even more valuable if the text can be made more concise, by identifying a limited number of key take-home messages of the study, and rewriting the text more straight-to-the-point to convey these few key messages to the reader. Additional material can be moved to a Supplement.***"**

**Response 1:**

We thank the reviewer for the positive assessment of our manuscript and for the suggestion to streamline the text and better highlight the key take-home messages.

We would like to note that the research questions are already clearly stated in the Introduction, and the main findings and take-home messages will be more clearly highlighted in the Conclusions section.

However, we acknowledge the importance of making these elements even more visible to the reader. To this end, we will revise the manuscript to ensure that the research objectives and core contributions stand out more clearly throughout the text. We will also relocate some descriptive or methodological details to the Supplementary Material.

**2. Specific comments**

**Comment 1:**

 *"The Authors must make sure that they consistently adhere to a single term to refer to a given concept. If different words are used interchangeably to denote the same idea, it creates confusion for the reader. Examples are as follows: - firms, enterprises ...: if they all mean the same, choose one word and stick to it; - local-scale (title), micro-scale, micro-local (Abstract) ... - "multiple regression model" "*.

**Response 1:**

We will standardise the vocabulary throughout the document. In detail, we will use the term "firms" to refer to economic entities and "micro-scale" to refer to the analysis at the individual level, avoiding ambiguities. The term "multiple regression model" will be standardised throughout the text.

**Comment 2:**

*"L52: "The assessment of damage is important to evaluate actions from ..." Is the model proposed by the Authors suitable for evaluating a wide range of risk reduction measures. It is important to discuss this point in Section 8, particularly in light of regression models which do not consider any hazard explanatory variable."*

**Response 2:**

We appreciate this observation. In our analysis, water depth (WD) is incorporated as an explanatory variable specifically for stock damage, where it is statistically significant. However, for total damage and the other components (structure and equipment), WD was not statistically significant—likely due to its limited variability in our dataset ( 50% of the data are

concentrated between 0.4 -1 m). On the other hand, the model includes the economic sector as an explanatory variable, which allows us to capture differences in sectoral vulnerability. This means that risk can be reduced not only by acting on the hazard reduction but also by addressing vulnerability through sector-specific strategies.

While the model is not suitable for assessing the effectiveness of site-specific or firm-level interventions, it is particularly well suited to support collective, sector-level risk management actions. These include the design of adaptation policies, targeted insurance schemes, and continuity planning for the most vulnerable types of businesses. In this sense, the model provides valuable insights for public authorities and policymakers aiming to reduce economic flood risk by acting on structural characteristics of vulnerability, rather than just on hazard mitigation.

In addition, the model offers an improvement over foreign models for estimating business damage in the Italian context, as evidenced by the reduction in average error compared to pre-existing approaches. We will expand the discussion in Section 8 to clarify that.

**Comment 3:**

*"L160-164: exactly how data was collected and how it was processed is key to the analysis. I recommend that a Supplement to the paper is created to provide more details on how data was collected and treated for each flood event."*

**Response 3:**

Thank you for this recommendation, we will add a description on how data was collected and treated for each event in the supplement.

**Comment 4:**

*"L170-174: exactly how water depth was estimated is crucial for the entire analysis. The Authors are asked to describe in detail how the hydrodynamic models were validated (which is their performance and hence estimate the uncertainty on the water depths), and how water depth attribution to (large) buildings was performed. The latter point refers to how a single representative value of water depth was assigned to a particular building from the gridded results of a 2D model. Considering the highest water depth around the building, or the average, or other statistics can lead to substantially different results, especially for large buildings. How were the buildings incorporated in the 2D simulations (building block approach vs. building hole approach ...)?"*

**Response 4:**

Thank you for this insightful comment. We agree that the methodology for estimating water depth is a crucial component in flood damage modelling. However, as this study is focused specifically on developing and validating an empirical damage model, a detailed analysis of the hydrodynamic simulations falls outside the scope of this work.

That said, we recognize the importance of model accuracy and have relied on previously validated hydraulic simulations from authoritative studies (Amadio et al., 2019; Scorzini et al., 2018; Carisi et al., 2018; Gatti, 2016), which provide a reliable basis for our damage analysis.

Regarding the assignment of water depth values to buildings, it is important to clarify that all analysed events are characterized by low-velocity, lowland floods. As such, water depth can be considered relatively uniform around individual buildings, even large ones. Therefore, the method of aggregating depth values (e.g., average vs. maximum) has negligible influence in this specific context.

We will add a clarifying sentence in Section 3 (L 175-176) to acknowledge this and to refer the reader explicitly to the original sources for details about model construction, calibration, and validation procedures.

**Comment 5:**

*"I'm particularly sceptical of all the boxplots presented with a log-scale y-axis extending up to 105 €/m² (Figure 1 and later). This is motivated by the existence of a few outliers; but I believe that this representation leads to a bias in the visual inspection of the plots. I would recommend keeping one version of the boxplots as they are, and including a second version based on a linear scale bounded by reasonable values (e.g., up to 5,000 €/m²). One version could remain in the main text, and the other one be moved to a Supplement."*

**Response 5:**

Thank you for this observation. The main reason why we choose this log-scale representation is because there is a very large variation in the data that is not possible to appreciate except on a logarithmic scale as you can see from below figures reporting reviewer's request. We think they are not really representative, but we may add them in the supplementary material if required. We are happy to follow the Editor's recommendation on whether to keep the log-scale plots in the main text or to replace them with the linear-scale version.

[Figure]

**Figure 1 :** Boxplots of the unitary total damage by economic category for the entire dataset (up to 5000 €/m²).

[Figure]

**Figure 2:** Boxplot of the unitary damage by component and by economic category (up to 5000 €/m²); (a) total damage ($d_t$) **(b)** damage to the structure ($d_{BS}$); **(c)** damage to the stock ($d_S$); **(d)** damage to the equipment ($d_E$).

**Comment 6:**

*"L265-267: this may be related to the technique used for water depth attribution to buildings from the 2D computations."*

**Response 6:**

We acknowledge the reviewer's observation. However, we believe that the limited statistical significance of water depth in our model is not primarily due to the method used for attributing water depth from 2D hydraulic simulations. The hydraulic models adopted in each case study were developed using state-of-the-art methods, and we consider them overall reliable. While some degree of uncertainty in the estimation of water depth is inevitable, we attribute the lack of statistical correlation mainly to a selection bias in the damage declaration dataset.

As discussed in Section 8 and Appendix A, there appears to be an underrepresentation of small enterprises affected by low water depths, which likely distorts the relationship between water depth and reported damage.

In addition, another potential source of uncertainty may lie in the estimation of surface area, which was not always available in the original dataset. In these cases, the surface was manually estimated through GIS analysis using topographic databases. While this allowed us to reconstruct missing values, it may have introduced further variability that interacts with water depth in affecting damage estimates. We will clarify this aspect more explicitly in the revised manuscript.

**Comment 7:**

*"284-286: this is not clear. Please reformulate and be sure to use the right words. Is "proportional" correct?"*

**Response 7:**

Thank you for your observation. We agree that the word was unclear, and we will revise the sentence for clarity. The term "proportional" was not appropriate, as the functional form adopted is exponential. What we intended to express is that variations in business size or water depth result in *multiplicative* effects on the estimated damage, consistent with the log-log model specification. We will reformulate the sentence accordingly in the revised manuscript.

**Comment 8:**

*"L299: what is meant with "some statistical measures". Be more specific. Which one is relevant here?"*

**Response 8:**

We will expand our explanation of the statistical advantages of log transformations, particularly, in handling skewed distributions. By "some statistical measures," we refer to indicators of central tendency and dispersion (such as the mean and standard deviation), which are less affected by extreme values when using log-transformed data. Log transformation also supports statistical inference by improving the performance of normality tests—such as the Shapiro-Wilk test—and helps satisfy Ordinary Least Squares (OLS) assumptions by stabilizing variance and reducing heteroskedasticity. Furthermore, it enhances the reliability of inferential statistics (e.g., the *t*-test), which rely on approximately normal distributions for valid significance testing. Measures of model performance, such as the adjusted $R^2$, may also improve due to better linearity and reduced sensitivity to outliers. Lastly, reducing skewness in independent variables contributes to more robust and interpretable coefficient estimates in regression models.

**Comment 9:**

*"L309 and Table 5: it is unclear how the results for model training and validation are displayed. Importantly, it is also not clear what is presented in Table 5 regarding the 10 folds ... Does Table 5 display an average over the 10 folds, or something else? This is key to clarify."*

**Response 9:**

Thank you for pointing this out. We will clarify that the results presented in Table 5 represent the average of the coefficients estimated across the 10 folds in the cross-validation procedure. Specifically, for each fold, the model was trained on 90% of the data and validated on the remaining 10%. The reported coefficients and goodness-of-fit metrics (e.g., adjusted $R^2$, RMSE, CV) are the average values computed over the 10 iterations. This approach was adopted to enhance the robustness of the model evaluation and reduce the influence of data partitioning.

**Comment 10:**

*"L310-315: it is unclear whether "damage" or "unitary damage" is considered. The text seems to lack rigor. Percentages are given; but it is not clear to what they refer (average value, median value …)."*

**Response 10:**

Thank you for the comment. We clarified in Line 225 that all analyses refer to unitary damage, defined as total reported damage divided by the surface of the economic activity. To enhance clarity and consistency, we will revise the relevant sections of the manuscript to ensure the term unitary damage is used uniformly throughout.

Regarding the percentage values mentioned (Lines 315–320), we confirm that these are derived from the estimated coefficients of the regression model with the logarithm of unitary damage as the dependent variable. In the case of dummy variables (e.g., sector categories), the interpretation of the estimated coefficient $\beta$ in a log-log model follows the standard transformation:

Percentage change = $100 \times (e^{\beta} - 1)$

For example, the coefficient for the manufacturing sector (Dc) is approximately 0.63, which implies an expected 88% higher unitary damage compared to the reference category (commercial sector). Conversely, the coefficient for the office category (Doff) implies a 67% lower unitary damage.

For continuous variables like surface area, the interpretation is in terms of elasticity: a 10% increase in firm size is associated with an approximate 8% decrease in unitary damage, based on a coefficient of –0.78.

We will revise the text to make these interpretations clearer.

**Comment 11:**

*" L316-318: elaborate a bit more on whether this result is satisfactory or not."*

**Response 11:**

Thank you for the suggestion. We agree that the interpretation of model performance is important. However, since Section 5 is focused on presenting the results, we have chosen to describe the outcomes without evaluating them in detail at that point. A more thorough assessment of the model's performance—including a discussion on the adequacy of the adjusted $R^2$, RMSE, and potential limitations—is provided in Section 8. We believe this structure helps maintain a clear separation between results and their critical interpretation.

**Comment 12:**

*"Table 7 could be moved to a Supplement and replaced by a bar chart in the main text. The last row of Table 7 is not clear "Total average"*

**Response 12:**

Thank you for the suggestion. While we acknowledge that a bar chart may enhance visual appeal, we believe that Table 7 provides a clearer and more precise representation of the average values across economic sectors and damage components. In particular, the tabular format allows for immediate readability and direct numerical comparison, which we consider important for the interpretation of the results.

For this reason, we would prefer to retain Table 7 in the main text, and possibly include a complementary bar chart in the Supplementary Material.

Additionally, we will revise the caption to clarify that the "Total average" refers to the overall average across all sectors.

**Comment 13:**

*"Paragraph starting in Line 365: the use of "initial value" is not clear and seems not rigorous."*

**Response 13:**

Thank you for your comment. We agree that the term "initial value" is unclear and potentially misleading. We will remove it from the paragraph to improve clarity and rigor.

**Comment 14:**

*"L375: this seems incorrect. It is a portion of the variance which is explained. Again, bi rigorous with the use of the words "damage" or "unitary damage".*

**Response 14:**

Thank you for pointing this out. We acknowledge the incorrect phrasing, and we will revise the sentence to clarify that it refers to the portion of variance explained by the model. We will precise use of the terms "damage" and "unit damage" throughout the text.

**Comment 15:**

*"It is essential that the performance of each "final" model (Eqs. 11 to 13) is represented by means of a scatter plot (computed vs. observed values), and that the possible trends revealed by these plots (over- vs. under-estimations ...) are briefly discussed."*

**Response 15:**

Thank you for the observation, the scatterplot will be added and will be briefly discussed.

**Comment 16:**

*"Lines 402 and following: it is not clear which are the "Italian models". Are there only two such models (Arrighi et al. and Molinari et al., 2020), or also others? Besides, make sure to avoid contradictions in the text: L469 it is stated that the new model applies to the "whole Italian territory". Then, also Florence? In that case, how does it compare with Arrighi et al.? It is probably better to tone down the statement "whole Italian territory" and emphasize the regions for which data were available and were used for calibrating the regressions."*

**Response 16:**

Thank you for this insightful comment. We acknowledge the need for greater clarity. The model by Arrighi et al. was specifically developed for the city of Florence and is based on local market values, particularly real estate prices. This makes direct comparison with our model difficult. In contrast, our model was calibrated using data from multiple regions across Italy, which enhances its applicability beyond a single local context.

**Comment 17:**

*"L410: discussion on relative vs. absolute errors is not clear. Please clarify."*

**Response 17:**

Thank you for your comment. We acknowledge the need for greater clarity. We will clarify the distinction between relative and absolute errors. Specifically, we will explain that, since the model is trained on the logarithm of the unitary damage, it effectively minimizes relative errors (i.e., proportional differences between predicted and observed values) rather than absolute errors (i.e., raw differences).

**Comment 18:**

*"The main results need to be concisely summarized in the Conclusion."*

**Response 18:**

Thank you for your suggestion. We will revise the Conclusion section to include a concise summary of the main results, highlighting the key findings of our analysis and their implications.

**3. Technical corrections**

**Comment 1:**

*"L22-23: rephrase to improve clarity."*

**Response 1:**

Thank you for your comment. We will rephrase lines 22–23 to improve clarity and ensure the sentence is more concise and accessible to the reader.

**Comment 2:**

*"L23: is "vulnerable" the correct word? Data show that these are the highest values of damage. It is not proven that the reason for the higher damage is a higher vulnerability. The hazard may be higher."*

**Response 2:**

Thank you for your insightful comment. We will change the misleading word in the new version with "susceptible to".

**Comment 3:**

*"Remove multiple occurrences of "in fact" throughout the text."*

Response 3:
We will remove all unnecessary occurrences of "in fact" throughout the text.

**Comment 4:**

*"Many references cited in the text are missing in the reference list."*

**Response 4:**
We will review and corrected the reference list to ensure all cited works are included.

**Comment 5**:

*"Table 2: bottom left cell should read 812 instead of 325."*

**Response 5:**

Thank you for noticing. We will correcte the value in the bottom-left cell of Table 2 to 812.

**Comment 6:**

*"L221: same approach as."*

**Response 6:**

Corrected the sentence in line 221 for clarity.

**Comment 7:**

*"In general, avoid repeating generally known information, such as a description of a boxplot (e.g., in caption of*

*Figure 1)."*

**Response 7:**

We will remove the generic description of the boxplot from the caption.

**Comment 8:**

*"There are errors in the numbering of the equations."*

**Response 8:**

Equation numbering will be reviewed and corrected.

**Comment 9:**

*"L293: "take unity values". "*

**Response 9:**

We will revise the expression "take unity values" for clarity.

**Comment 10:**

*"L299: remove "caused" "*

**Response 10:**

We will remove the word "caused" as suggested.

**Comment 11:**

*"In figures like Figure 7, the exact meaning of the grey shaded area needs to be explained."*

**Response 11:**

We will add a clarification in the caption to explain the grey shaded area in Figure 7.

**Comment 12:**

*"L387. Reformulate "inadvisable"."*

**Response 12:**

We will reformulate the sentence to replace "inadvisable" with a clearer term.

**Comment 13:**

*" L391: remove "at the premises' location"."*

**Response 13:**

We will remove the phrase "at the premises' location" as recommended.

**Comment 14:**

*"L463. Delete the first sentence ("… results were predictable.")."*

**Response 14:**

We will delete the first sentence in line 463 as suggested.

---

## Author Comment (AC2)

**Econometric Modelling for Estimating Direct Flood Damage to Firms: A Local-Scale Approach Using Post-Event Records in Italy.**

*Ballocci M., Molinari D., Marin G., Galliani M., Domeneghetti A., Menduni G., Sterlacchini S., and Ballio F.*

**Response to the Reviewers' comments**

We thank the Editors and the Reviewer for their interest in our manuscript and the valuable comments they have provided. According to them, the  manuscript will be revised as follow.

In the following a point-by-point answer to reviewers' comment is provided.

**Reviewer #2**

**1. General comments**

**Comment 1:**

*"While the study outlines in a clear way the econometric modelling approach, additional details on data collection, model specification, and variable selection process would enhance reproducibility and understanding."*

**Response 1:**

We thank the reviewer for this valuable comment. We appreciate the importance of transparency in econometric research and have revised the manuscript to better clarify key aspects related to data collection, model specification, and variable selection.

***Data collection:***

In response to this point, we will add a dedicated subsection titled "4. Detailed Data Collection" in the Supplement, which provides further explanation of the post-event survey methodology, the format of the forms used, the institutions responsible for their compilation (e.g. municipalities, regions), and the standardization procedure adopted prior to the analysis. This addition aims to increase the reproducibility of our data processing steps and enhance clarity on the original source and structure of the information.

***Model Specification:***

The main manuscript already presents the econometric model in detail in Section 4 (Method), including the log-log transformation of variables, the rationale for this choice (e.g., to reduce skewness and heteroskedasticity), and the regression equation used to estimate the physical damage to assets. However, it is not entirely clear to us what specific aspects of the model the reviewer would like to see further detailed or clarified.

***Variable Selection Process:***

The variables included in the model were selected based on both theoretical considerations and empirical evidence from the existing literature (e.g., Merz et al., 2010; Paprotny et al., 2020) and as well as on the availability of data, as already discussed in Section 3 (Data). Specifically, the choice of water depth, surface, and economic activity type reflects their well-established relevance in flood damage modelling.

**Comment 2:**

*"The manuscript would benefit from a more in-depth discussion of the limitations associated with the dataset, such as potential biases in post-event data collection, missing data, and the representativeness of the sample."*

**Response 2:**

We fully agree with the reviewer on the importance of acknowledging the limitations of the dataset used in our study, and we will revise the manuscript accordingly.

A dedicated paragraph will be added to the Discussion section to elaborate on three key aspects.

*Post-Event Data Collection Bias:*

As the dataset relies on self-reported information collected shortly after flood events, it is subject to potential inaccuracies due to memory recall, subjective estimations of damages by business owners, or the pressure to report higher damages to access public compensation schemes. We acknowledge this as a source of uncertainty, and we will explicitly discuss it in the revised text.

*Missing Data:*

The only missing information in the dataset concerned the surface area of the building. In these cases, we estimated the area using regional topographic databases, which may have introduced some uncertainty in the values.

*Sample Representativeness:*

Our sample covers five flood events in Italy. However, these cases were selected based on data availability and quality, and do not represent a statistically random sample of all flood-affected firms in the country. The representativeness of the sample (issue discussed in **Appendix – Section 4**)  is limited not only by the selection of the case studies but also by the characteristics of the firms that responded. It is possible that firms more severely affected, or more structured and proactive in damage reporting, are overrepresented and vice versa. Therefore, while the data provide valuable micro-scale insights, we caution against generalizing the findings to all flood-affected firms in Italy or other contexts.

**Specific comments**

**Comment 1:**

*"L. 14 and throughout the text: ensure consistent use of terms such as "firms," "enterprises," and "businesses" throughout the manuscript to avoid confusion."*

**Response 1:**

We will standardise the terminology throughout the manuscript. From now on, we will consistently use the term "firms" to refer to economic entities.

**Comment 2:**

"L. 17: add "of" after understanding."

**Response2:**

The preposition "of" will be added after "understanding" to correct the sentence.

**Comment 3:**

*"L. 34-35: I suggest not starting the introduction by stating what the manuscript is about. The objectives of the study should be presented at the end of the introduction."*

**Response 3:**

We appreciate the reviewer's stylistic suggestion. However, we deliberately chose to introduce the topic by briefly stating the focus of the manuscript in the opening lines, as we believe it is effective to immediately convey the scope and relevance of the work, without requiring the reader to wait until the end of the introduction to understand what the paper is about.

That said, we agree that the research objectives should be explicitly stated, and we will ensure they are clearly formulated in the final paragraph of the Introduction.

**Comment 4:**

*"L. 60: extant->existing"*

**Response 4:**

We will replace "extant" with "existing".

**Comment 5:**

*"L. 160-174: more details on data collection and water depth estimation are fundamental for a better understanding of the analysis."*

**Response 5:**

We thank the reviewer for this valuable comment. While we fully acknowledge the relevance of providing further technical details on data collection and water depth estimation, we have intentionally chosen to focus this paper on the econometric modelling of flood damage. Including an in-depth discussion of hydraulic modelling would have extended the scope beyond the intended objective of presenting a concise and targeted contribution.

Nonetheless, we have already included references to the relevant technical studies (Amadio et al., 2019; Scorzini et al., 2018; Carisi et al., 2018; Gatti, 2016) that describe the hydraulic modelling procedures and water depth reconstruction used in the case studies. Interested readers can refer to these sources for a detailed explanation of the flood modelling framework.

**Comment 6:**

*"L. 209: from which Topographic Databases? More details are needed here."*

**Response 6:**

Thank you for this observation. We will clarify in the revised manuscript that the Topographic Databases used to localize the economic assets refer to the official regional Topographic Databases (DBT) provided by public authorities (e.g., Regione Lombardia, Regione Sardegna).

**Comment 7:**

*"References: check the reference list carefully, as in some cases I couldn't find the article cited in the text."*

**Response 7:**

We thank the reviewer for this observation. We will carefully review the manuscript and the reference list to ensure consistency between in-text citations and listed references. Any missing or incorrect entries will be corrected. All cited sources will properly include in the reference list.

---

## Author Response (AR1)

**Econometric Modelling for Estimating Direct Flood Damage to Firms: A Local-Scale Approach Using Post-Event Records in Italy.**

*Ballocci M., Molinari D., Marin G., Galliani M., Domeneghetti A., Menduni G., Sterlacchini S., and Ballio F.*

**Response to the Reviewers' comments**

We thank the Editors and the Reviewers for their interest in our manuscript and the valuable comments they have provided which contributed to increase the paper quality and robustness.

A point-by-point response to the reviewers' comments is provided below. Lines number refers to the marked version of the revised manuscript

**Reviewer #1**

**1. General Comments**

"*My opinion, is that the paper is suitable for publication in this journal, provided the points below are addressed, but it would become even more valuable if the text can be made more concise, by identifying a limited number of key take-home messages of the study, and rewriting the text more straight-to-the-point to convey these few key messages to the reader. Additional material can be moved to a Supplement.*"

**Response 1:**

We thank the reviewer for the positive assessment of our manuscript and for the suggestion to streamline the text and better highlight the key take-home messages.

In the new version, we highlighted more clearly the main findings and take-home messages in the Conclusions section (L518-522).

Moreover, to make the paper more concise, we have relocated some methodological details to the Supplementary Material.

**2. Specific comments**

**Comment 1:**

"*The Authors must make sure that they consistently adhere to a single term to refer to a given concept. If different words are used interchangeably to denote the same idea, it creates confusion for the reader. Examples are as follows: - firms, enterprises ...: if they all mean the same, choose one word and stick to it; - local-scale (title), micro-scale, micro-local (Abstract) ... - "multiple regression model"*".

**Response 1:**

We standardised the vocabulary throughout the document. In detail, we used the term "firms" to refer to economic entities and "micro-scale" to refer to the analysis at the individual level, avoiding ambiguities. The term "multiple regression model" have been standardised throughout the text.

**Comment 2:**

"*L52: "The assessment of damage is important to evaluate actions from ..." Is the model proposed by the Authors suitable for evaluating a wide range of risk reduction measures. It is important to discuss this point in Section 8, particularly in light of regression models which do not consider any hazard explanatory variable.*"

**Response 2:**

We appreciate this observation. In our analysis, water depth (WD) is incorporated as an explanatory variable specifically for stock damage, where it is statistically significant. However, for total damage and the other components (structure and equipment), WD was not statistically significant—likely due to its limited variability in our dataset ( 50% of the data are concentrated between 0.4 -1 m). On the other hand, the model includes the economic sector as an explanatory variable,

which allows us to capture differences in sectoral vulnerability. This means that risk can be reduced not only by acting on the hazard reduction but also by addressing vulnerability through sector-specific strategies.

While the model is not suitable for assessing the effectiveness of site-specific or firm-level interventions, it is particularly well suited to support collective, sector-level risk management actions. These include the design of adaptation policies, targeted insurance schemes, and continuity planning for the most vulnerable types of businesses. In this sense, the model provides valuable insights for public authorities and policymakers aiming to reduce economic flood risk by acting on structural characteristics of vulnerability, rather than just on hazard mitigation.

In addition, the model offers an improvement over foreign models for estimating business damage in the Italian context, as evidenced by the reduction in average error compared to pre-existing approaches. We have expanded the discussion in Section 8 to clarify that (L503-514).

**Comment 3:**

*"L160-164: exactly how data was collected and how it was processed is key to the analysis. I recommend that a Supplement to the paper is created to provide more details on how data was collected and treated for each flood event."*

**Response 3:**

Thank you for this recommendation, we added a description on how data was collected and treated for each event in the supplement.

**Comment 4:**

*"L170-174: exactly how water depth was estimated is crucial for the entire analysis. The Authors are asked to describe in detail how the hydrodynamic models were validated (which is their performance and hence estimate the uncertainty on the water depths), and how water depth attribution to (large) buildings was performed. The latter point refers to how a single representative value of water depth was assigned to a particular building from the gridded results of a 2D model. Considering the highest water depth around the building, or the average, or other statistics can lead to substantially different results, especially for large buildings. How were the buildings incorporated in the 2D simulations (building block approach vs. building hole approach ...)?"*

**Response 4:**

Thank you for this insightful comment. We agree that the methodology for estimating water depth is a crucial component in flood damage modelling. However, as this study is focused specifically on developing and validating an empirical damage model, we believe a detailed analysis of the hydrodynamic simulations falls outside the scope of this work.

That said, we recognize the importance of model accuracy and have relied on previously validated hydraulic simulations from authoritative studies (Amadio et al., 2019; Scorzini et al., 2018; Carisi et al., 2018; Gatti, 2016), which provide a reliable basis for our damage analysis.

Regarding the assignment of water depth values to buildings, it is important to clarify that all analysed events are characterized by low-velocity, riverine floods on relatively flat areas. As such, water depth can be considered relatively uniform around individual buildings, even large ones. Therefore, the method of aggregating depth values (e.g., average vs. maximum) has negligible influence in this specific context.

We added a clarifying sentence in Section 3 (L174-175) to acknowledge this and to refer the reader explicitly to the original sources for details about model construction, calibration, and validation procedures.

**Comment 5:**

*"I'm particularly sceptical of all the boxplots presented with a log-scale y-axis extending up to 105 €/m² (Figure 1 and later). This is motivated by the existence of a few outliers; but I believe that this representation leads to a bias in the visual inspection of the plots. I would recommend keeping one version of the boxplots as they are and including a second version based on a linear scale bounded by reasonable values (e.g., up to 5,000 €/m²). One version could remain in the main text, and the other one be moved to a Supplement."*

**Response 5:**

Thank you for this observation. The main reason why we choose the log-scale representation is because there is a very large variation in the data that is not possible to appreciate except with a logarithmic scale as you can see from below Figure 1 and Figure 2 reporting reviewer's request. We think they are not really explicative of the data, but we may add them in the supplementary material if required. We are happy to follow the Editor's recommendation on this point.

[Figure]

**Figure 1:** Boxplots of the unitary total damage by economic category for the entire dataset (up to 5000 €/m$^2$).

[Figure]

**Figure 2:** Boxplot of the unitary damage by component and by economic category (up to 5000 €/m$^2$); **(a)** total damage ($d_t$); **(b)** damage to the structure ($d_{BS}$); **(c)** damage to the stock ($d_S$); **(d)** damage to the equipment ($d_E$).

**Comment 6:**

*"L265-267: this may be related to the technique used for water depth attribution to buildings from the 2D computations."*

**Response 6:**

We acknowledge the reviewer's observation. However, we believe that the limited statistical significance of water depth in our model is not primarily due to the method used for attributing water depth from 2D hydraulic simulations. The hydraulic models adopted in each case study were developed using state-of-the-art methods, and we consider them overall reliable. While some degree of uncertainty in the estimation of water depth is inevitable, we attribute the lack of statistical correlation mainly to a selection bias in the damage declaration dataset.

As discussed in Section 8 and Appendix A, there appears to be an underrepresentation of small enterprises affected by low water depths, which likely distorts the relationship between water depth and reported damage.

In addition, another potential source of uncertainty may lie in the estimation of surface area, which was not always available in the original dataset. In these cases, the surface was manually estimated through GIS using topographic databases. While this allowed us to reconstruct missing values, it may have introduced further variability that interacts with water depth in affecting damage estimates. We clarified this aspect more explicitly in the revised manuscript. (L482 – 486)

**Comment 7:**

*"284-286: this is not clear. Please reformulate and be sure to use the right words. Is "proportional" correct?"*

**Response 7:**

Thank you for your observation. We agree that the word was unclear and revised the sentence for clarity. The term "proportional" was not appropriate, as the functional form adopted is exponential. What we intended to express is that variations in business size or water depth result in *multiplicative* effects on the estimated damage, consistent with the log-log model specification. We have reformulated the sentence accordingly in the revised manuscript.

**Comment 8:**

*"L299: what is meant with "some statistical measures". Be more specific. Which one is relevant here?"*

**Response 8:**

We expanded our explanation of the statistical advantages (L303) of log transformations, particularly, in handling skewed distributions. By "some statistical measures," we refer to indicators of central tendency and dispersion (such as the mean and standard deviation), which are less affected by extreme values when using log-transformed data. Log transformation also supports statistical inference by improving the performance of normality tests—such as the Shapiro-Wilk test—and helps satisfy Ordinary Least Squares (OLS) assumptions by stabilizing variance and reducing heteroskedasticity. Furthermore, it enhances the reliability of inferential statistics (e.g., the *t*-test), which rely on approximately normal distributions for valid significance testing. Measures of model performance, such as the adjusted $R^2$, may also improve due to better linearity and reduced sensitivity to outliers. Lastly, reducing skewness in independent variables contributes to more robust and interpretable coefficient estimates in regression models.

**Comment 9:**

*"L309 and Table 5: it is unclear how the results for model training and validation are displayed. Importantly, it is also not clear what is presented in Table 5 regarding the 10 folds ... Does Table 5 display an average over the 10 folds, or something else? This is key to clarify."*

**Response 9:**

Thank you for pointing this out. We clarified (L314 - 315) that the results presented in Table 5 represent the average of the coefficients estimated across the 10 folds in the cross-validation procedure. Specifically, for each fold, the model was trained on 90% of the data and validated on the remaining 10%. The reported coefficients and goodness-of-fit metrics (e.g., adjusted $R^2$, RMSE, CV) are the average values computed over the 10 iterations. This approach was adopted to enhance the robustness of the model evaluation and reduce the influence of data partitioning.

**Comment 10:**

*"L310-315: it is unclear whether "damage" or "unitary damage" is considered. The text seems to lack rigor. Percentages are given; but it is not clear to what they refer (average value, median value ...)."*

**Response 10:**

Thank you for the comment. We clarified in Line 225 that all analyses refer to unitary damage, defined as total reported damage divided by the surface of the economic activity. To enhance clarity and consistency, we revised the relevant sections of the manuscript to ensure the term unitary damage is used uniformly throughout.

Regarding the percentage values mentioned (Lines 321–323), we confirm that these are derived from the estimated coefficients of the regression model with the logarithm of unitary damage as the dependent variable. In the case of dummy variables (e.g., sector categories), the interpretation of the estimated coefficient $\beta$ in a log-log model follows the standard transformation:

Percentage change = $100 \times (e^{\beta} - 1)$

For example, the coefficient for the manufacturing sector (Dc) is approximately 0.63, which implies an expected 88% higher unitary damage compared to the reference category (commercial sector). Conversely, the coefficient for the office category (Doff) implies a 67% lower unitary damage.

For continuous variables like surface area, the interpretation is in terms of elasticity: a 10% increase in firm size is associated with an approximate 8% decrease in unitary damage, based on a coefficient of –0.78.

We added footnote 3 to make these interpretations clearer.

**Comment 11:**

*" L316-318: elaborate a bit more on whether this result is satisfactory or not."*

**Response 11:**

Thank you for the suggestion. We agree that the interpretation of model performance is important. However, Section 5 is focused on presenting the results, without their discussion. A more thorough assessment of the model's performance— including a discussion on the adequacy of the adjusted $R^2$, RMSE, and potential limitations—is provided in Section 8. We believe this structure of the manuscript helps maintain a clear separation between results and their critical interpretation.

**Comment 12:**

*"Table 7 could be moved to a Supplement and replaced by a bar chart in the main text. The last row of Table 7 is not clear "Total average"*

**Response 12:**

Thank you for the suggestion. While we acknowledge that a bar chart may enhance visual appeal, we believe that Table 7 provides a clearer and more precise representation of the average values across economic sectors and damage components. In particular, the tabular format allows for immediate readability and direct numerical comparison, which we consider important for the interpretation of the results.

For this reason, we would prefer to retain Table 7 in the main text and possibly include a complementary bar chart in the Supplementary Material.

Additionally, we revised the caption to clarify that the "Total average" refers to the overall average across all sectors.

**Comment 13:**

*"Paragraph starting in Line 365: the use of "initial value" is not clear and seems not rigorous."*

**Response 13:**

Thank you for your comment. We agree that the term "initial value" is unclear and potentially misleading. We removed it from the paragraph to improve clarity and rigor.

**Comment 14:**

*"L375: this seems incorrect. It is a portion of the variance which is explained. Again, bi rigorous with the use of the words "damage" or "unitary damage".*

**Response 14:**

Thank you for pointing this out. We acknowledge the incorrect phrasing, and we revised the sentence to clarify that it refers to the portion of variance explained by the model. We have clarified the use of the terms "damage" and "unit damage" throughout the text.

**Comment 15:**

*"It is essential that the performance of each "final" model (Eqs. 11 to 13) is represented by means of a scatter plot (computed vs. observed values), and that the possible trends revealed by these plots (over- vs. under-estimations ...) are briefly discussed."*

**Response 15:**

Thank you for the observation, the scatterplots have been added and briefly discussed (Fig.5; L334-337 and Fig. 9; L403-45).

**Comment 16:**

*"Lines 402 and following: it is not clear which are the "Italian models". Are there only two such models (Arrighi et al. and Molinari et al., 2020), or also others? Besides, make sure to avoid contradictions in the text: L469 it is stated that the new model applies to the "whole Italian territory". Then, also Florence? In that case, how does it compare with Arrighi et al.? It is probably better to tone down the statement "whole Italian territory" and emphasize the regions for which data were available and were used for calibrating the regressions."*

**Response 16:**

Thank you for this insightful comment. We acknowledge the need for greater clarity. The model by Arrighi et al. was specifically developed for the city of Florence and is based on local market values, particularly real estate prices. This makes direct comparison with our model difficult although the proposed model can be applied to Florence too. In contrast, our model was calibrated using data from multiple regions across Italy, which enhances its applicability beyond a single local context.

**Comment 17:**

*"L410: discussion on relative vs. absolute errors is not clear. Please clarify."*

**Response 17:**

Thank you for your comment. We acknowledge the need for greater clarity. We clarified (L431-432) the distinction between relative and absolute errors. Specifically, we have explained that, since the model is trained on the logarithm of the unitary damage, it effectively minimizes relative errors (i.e., proportional differences between predicted and observed values) rather than absolute errors (i.e., raw differences).

**Comment 18:**

*"The main results need to be concisely summarized in the Conclusion."*

**Response 18:**

Thank you for your suggestion. We revised the Conclusion section to include a concise summary of the main results, highlighting the key findings of our analysis and their implications.

**3. Technical corrections**

**Comment 1:**

*"L22-23: rephrase to improve clarity."*

**Response 1:**

Thank you for your comment. We rephrased lines 22–23 to improve clarity and ensure the sentence is more concise and accessible to the reader.

**Comment 2:**

*"L23: is "vulnerable" the correct word? Data show that these are the highest values of damage. It is not proven that the reason for the higher damage is a higher vulnerability. The hazard may be higher."*

**Response 2:**

Thank you for your insightful comment. We rephrased lines 22–23 to improve clarity and ensure the sentence is more concise and accessible to the reader. ".

**Comment 3:**

*"Remove multiple occurrences of "in fact" throughout the text."*

**Response 3:**
We removed all unnecessary occurrences of "in fact" throughout the text.

**Comment 4:**

*"Many references cited in the text are missing in the reference list."*

**Response 4:**
We reviewed and corrected the reference list to ensure all cited works are included.

**Comment 5**:

*"Table 2: bottom left cell should read 812 instead of 325."*

**Response 5:**

Thank you for noticing. We corrected the value in the bottom-left cell of Table 2 to 812.

**Comment 6:**

*"L221: same approach as."*

**Response 6:**

Corrected the sentence in line 221 for clarity.

**Comment 7:**

*"In general, avoid repeating generally known information, such as a description of a boxplot (e.g., in caption of*

*Figure 1)."*

**Response 7:**

 We removed the generic description of the boxplot from the caption.

**Comment 8:**

*"There are errors in the numbering of the equations."*

**Response 8:**

Equation numbering has been reviewed and corrected.

**Comment 9:**

*"L293: "take unity values". "*

**Response 9:**

We revised the expression "take unity values" for clarity.

**Comment 10:**

*"L299: remove "caused" "*

**Response 10:**

We removed the word "caused" as suggested.

**Comment 11:**

*"In figures like Figure 7, the exact meaning of the grey shaded area needs to be explained."*

**Response 11:**

We added a clarification in the caption to explain the grey shaded area in Figure 7.

**Comment 12:**

*"L387. Reformulate "inadvisable"."*

**Response 12:**

We reformulated the sentence to replace "inadvisable" with a clearer term.

**Comment 13:**

*" L391: remove "at the premises' location"."*

**Response 13:**

We removed the phrase "at the premises' location" as recommended.

**Comment 14:**

*"L463. Delete the first sentence ("… results were predictable.")."*

**Response 14:**

We deleted the first sentence in line 463 as suggested.

**Reviewer #2**

**1. General comments**

**Comment 1:**

*"While the study outlines in a clear way the econometric modelling approach, additional details on data collection, model specification, and variable selection process would enhance reproducibility and understanding."*

**Response 1:**

We thank the reviewer for this valuable comment. We appreciate the importance of transparency in econometric research and have revised the manuscript to better clarify key aspects related to data collection, model specification, and variable selection.

**Data collection:**

In response to this point, we added a dedicated subsection titled "Detailed Data Collection" in the Supplement, which provides further explanation of the post-event survey methodology, the format of the forms used, the institutions responsible for their compilation (e.g. municipalities, regions), and the standardization procedure adopted prior to the analysis. This addition aims to increase the reproducibility of our data processing steps and enhance clarity on the original source and structure of the information.

**Model Specification:**

The main manuscript already presents the econometric model in detail in Section 4 (Method), including the log-log transformation of variables, the rationale for this choice (e.g., to reduce skewness and heteroskedasticity), and the regression equation used to estimate the physical damage to assets. However, it is not entirely clear to us what specific aspects of the model the reviewer would like to see further detailed or clarified.

**Variable Selection Process:**

The variables included in the model were selected based on both theoretical considerations and empirical evidence from the existing literature (e.g., Merz et al., 2010; Paprotny et al., 2020) and as well as on the availability of data, as already discussed in Section 3 (Data). Specifically, the choice of water depth, surface, and economic activity type reflects their well-established relevance in flood damage modelling.

**Comment 2:**

*"The manuscript would benefit from a more in-depth discussion of the limitations associated with the dataset, such as potential biases in post-event data collection, missing data, and the representativeness of the sample."*

**Response 2:**

We fully agree with the reviewer on the importance of acknowledging the limitations of the dataset used in our study, and we revised the manuscript accordingly. We elaborated more on this aspect in the Discussion section (L480-483).

**Post-Event Data Collection Bias:**

As the dataset relies on self-reported information collected shortly after flood events, it is subject to potential inaccuracies due to memory recall, subjective estimations of damages by business owners, or the pressure to report higher damages to access public compensation schemes. We acknowledge this as a source of uncertainty, and we explicitly discussed it in the revised text.

**Missing Data:**

The only missing information in the dataset concerned the surface area of the building. In these cases, we estimated the area using regional topographic databases, which may have introduced some uncertainty in the values (L481 – 485).

*Sample Representativeness:*

Our sample covers five flood events in Italy. However, these cases were selected based on data availability and quality, and do not represent a statistically random sample of all flood-affected firms in the country. The representativeness of the sample (issue discussed in **Appendix – "Selection Bias"**) is limited not only by the selection of the case studies but also by the characteristics of the firms that responded. It is possible that firms more severely affected, or more structured and proactive in damage reporting, are overrepresented and vice versa. Therefore, while the data provide valuable micro-scale insights, we caution against generalizing the findings to all flood-affected firms in Italy or other contexts (L. 509-510).

**Specific comments**

**Comment 1:**

*"L. 14 and throughout the text: ensure consistent use of terms such as "firms," "enterprises," and "businesses" throughout the manuscript to avoid confusion."*

**Response 1:**

We standardised the terminology throughout the manuscript. From now on, we consistently used the term "firms" to refer to economic entities.

**Comment 2:**

"L. 17: add "of" after understanding."

**Response2:**

The preposition "of" has been added after "understanding" to correct the sentence.

**Comment 3:**

*"L. 34-35: I suggest not starting the introduction by stating what the manuscript is about. The objectives of the study should be presented at the end of the introduction."*

**Response 3:**

We appreciate the reviewer's stylistic suggestion. However, we deliberately chose to introduce the topic by briefly stating the focus of the manuscript in the opening lines, as we believe it is effective to immediately convey the scope and relevance of the work, without requiring the reader to wait until the end of the introduction to understand what the paper is about.

That said, we agree that the research objectives should be explicitly stated, and we ensured they are clearly formulated in the final paragraph of the Introduction.

**Comment 4:**

*"L. 60: extant->existing"*

**Response 4:**

We replaced "extant" with "existing".

**Comment 5:**

*"L. 160-174: more details on data collection and water depth estimation are fundamental for a better understanding of the analysis."*

**Response 5:**

We thank the reviewer for this valuable comment. While we fully acknowledge the relevance of providing further technical details on data collection and water depth estimation, we have intentionally chosen to focus this paper on the econometric modelling of flood damage. Including an in-depth discussion of hydraulic modelling would have extended the scope beyond the intended objective of presenting a concise and targeted contribution.

Nonetheless, we have already included references to the relevant technical studies (Amadio et al., 2019; Scorzini et al., 2018; Carisi et al., 2018; Gatti, 2016) that describe the hydraulic modelling procedures and water depth reconstruction used in the case studies. Interested readers can refer to these sources for a detailed explanation of the flood modelling framework.

**Comment 6:**

*"L. 209: from which Topographic Databases? More details are needed here."*

**Response 6:**

Thank you for this observation. We clarified in the revised manuscript that the Topographic Databases used to localize the economic assets refer to the official regional Topographic Databases (DBT) provided by public authorities (e.g., Regione Lombardia, Regione Sardegna), ( L216).

**Comment 7:**

*"References: check the reference list carefully, as in some cases I couldn't find the article cited in the text."*

**Response 7:**

We thank the reviewer for this observation. We carefully reviewed the manuscript and the reference list to ensure consistency between in-text citations and listed references. Any missing or incorrect entries will be corrected. All cited sources will properly include in the reference list.